# Kernel Stein Discrepancy thinning: a theoretical perspective of pathologies and a practical fix with regularization

**Clément Bénard**[1]       **Brian Staber**[1]       **Sébastien Da Veiga**[2]

[1] Safran Tech, Digital Sciences & Technologies, 78114 Magny-Les-Hameaux, France
[2] ENSAI, CREST, F-35000 Rennes, France
`{clement.benard, brian.staber}@{safrangroup.com}`
`sebastien.da-veiga@ensai.fr`

## Abstract

Stein thinning is a promising algorithm proposed by Riabiz et al. [2022] for post-processing outputs of Markov chain Monte Carlo (MCMC). The main principle is to greedily minimize the kernelized Stein discrepancy (KSD), which only requires the gradient of the log-target distribution, and is thus well-suited for Bayesian inference. The main advantages of Stein thinning are the automatic remove of the burn-in period, the correction of the bias introduced by recent MCMC algorithms, and the asymptotic properties of convergence towards the target distribution. Nevertheless, Stein thinning suffers from several empirical pathologies, which may result in poor approximations, as observed in the literature. In this article, we conduct a theoretical analysis of these pathologies, to clearly identify the mechanisms at stake, and suggest improved strategies. Then, we introduce the regularized Stein thinning algorithm to alleviate the identified pathologies. Finally, theoretical guarantees and extensive experiments show the high efficiency of the proposed algorithm. An implementation of regularized Stein thinning as the `kernax` library in python and JAX is available at https://gitlab.com/drti/kernax.

## 1   Introduction

Bayesian inference is a powerful approach to solve statistical tasks, and is especially efficient to incorporate prior expert knowledge of the studied system, or to provide uncertainties of the estimated quantities. Bayesian methods have thus demonstrated a high empirical performance for a wide range of applications, in particular in the fields of physics and computational biology, to just name a few. However, the Bayesian framework often leads to the evaluation of expectations with respect to a posterior distribution, which is not tractable [Green et al., 2015], except in the specific case of conjugate prior distribution and likelihood, which hardly occurs in practice. To overcome this issue, Markov chain Monte Carlo (MCMC) is one of the most commonly used computational methods to estimate these integrals. Indeed, MCMC algorithms iteratively generate a sample, which follows the targeted posterior distribution, as the Markov chain converges to its stationary state [Robert and Casella, 1999, Brooks et al., 2011]. Consequently, the quality of the resulting estimates strongly depends on the convergence of the MCMC and how its output is post-processed. Standard post-processing procedures of MCMC outputs consist in removing the first iterations, called the burn-in period, and thinning the Markov chain with a constant frequency. Burn-in removal aims at reducing the bias introduced by the random initialization of the Markov chain. The $\hat{R}$ convergence diagnosis of Gelman et al. [1995] is, for instance, a well known method for determining the burn-in period. On the other hand, thinning the Markov chain allows for compressing the MCMC output and may also reduce the correlation between the iteratively selected points. More recently, promising kernel-based

procedures were proposed to automatically remove the burn-in period, compress the output, and reduce the asymptotic bias [South et al., 2022]. These approaches consist in minimizing a kernel-based discrepancy measure $D(\mathbb{P}, \mathbb{Q}_m)$ between the empirical distribution $\mathbb{Q}_m$ of a subsample of the MCMC output of size $m$, and the target distribution $\mathbb{P}$. In this respect, minimization of the maximum mean discrepancy (MMD) was investigated by several authors, but these strategies require the full knowledge of the target distribution $\mathbb{P}$, whose density is not tractable in non-conjugate Bayesian inference.

Based on the previous works of Chen et al. [2018] and Chen et al. [2019], Riabiz et al. [2022] propose to minimize the kernelized Stein discrepancy (KSD), to design an efficient kernel-based algorithm to thin MCMC outputs in a non-tractable Bayesian setting. The KSD [Liu et al., 2016] is a score-based discrepancy measure, *i.e.*, it only requires the knowledge of the score function of the target $\mathbb{P}$, which is readily available in our Bayesian framework. Importantly, Gorham and Mackey [2017] showed that under suitable mild conditions, the KSD enjoys good convergence properties. More precisely, the KSD is a valid distance to detect samples drawn form the target distribution, provided that the sample size is large enough. Therefore, KSD thinning is a highly promising tool for post-processing and measuring the quality of MCMC outputs. This article thus focuses on the Stein thinning algorithm proposed by Riabiz et al. [2022], which consists in selecting $m$ points amongst the $n$ iterations of the MCMC output, by greedily minimizing the KSD distance. Thanks to the convergence properties of the KSD, the empirical measure of the selected points weakly converges towards the posterior law $\mathbb{P}$. However, on the practical side, several articles [Wenliang and Kanagawa, 2020, Korba et al., 2021] have noticed empirical limitations of KSD-based sampling algorithms, especially for multimodal target distributions. In fact, these limitations happen to be quite problematic, even in simple experiments, and have been slightly overlooked in the literature so far, in our opinion. Therefore, this article first focuses on the analysis of KSD pathologies in Section 2, taking both an empirical and theoretical point of view. Then, we propose strategies to mitigate the identified problems, and introduce the regularized Stein thinning in Section 3. We show the efficiency of our algorithm through both a theoretical analysis and extensive experiments in Section 4. Notice that proofs and additional experiments are gathered in Appendices 1-7 in the Supplementary Material. In the remaining of this initial section, we mathematically formalize the KSD distance and the associated Stein thinning algorithm.

**Kernelized Stein discrepancy.** Kernelized Stein discrepancy was independently introduced by Chwialkowski et al. [2016], Liu et al. [2016], Gorham and Mackey [2017] as a promising tool for measuring dissimilarities between two distributions $\mathbb{P}$ and $\mathbb{Q}$ on $\mathbb{R}^d$ with $d \geq 1$, whenever $\mathbb{P}$ admits a continuously differentiable density $p$, and the normalization constant of $p$ is not tractable. Let $k : \mathbb{R}^d \times \mathbb{R}^d \to \mathbb{R}$ be a positive semi-definite kernel and let $\mathcal{H}(k)$ be the associated reproducing kernel Hilbert space (RKHS) with inner product $\langle \cdot, \cdot \rangle_{\mathcal{H}(k)}$ and norm $\| \cdot \|_{\mathcal{H}(k)}$. Kernelized Stein discrepancy belongs to the family of maximum mean discrepancies (MMD) [Gretton et al., 2006] defined as

$$\mathrm{MMD}_k(\mathbb{P}, \mathbb{Q}) = \sup_{\|f\|_{\mathcal{H}(k)} \leq 1} |\mathbb{E}[f(\mathbf{X})] - \mathbb{E}[f(\mathbf{Z})]|, \tag{1}$$

where $\mathbf{X} \sim \mathbb{P}$, $\mathbf{Z} \sim \mathbb{Q}$. If the kernel $k$ is characteristic, then the MMD is a distance between probability distributions. In practice, the MMD may not be computable as it involves mathematical expectations with respect to $\mathbb{P}$, whose density is not tractable. To circumvent this issue, Gorham and Mackey [2015] proposed the Stein discrepancy which relies on Stein's method [Stein, 1972]. It consists in defining an operator $\mathcal{T}_p$ that maps functions $g : \mathbb{R}^d \to \mathbb{R}^d$ to real-valued functions such that $\mathbb{E}[\mathcal{T}_p g(\mathbf{X})] = 0$, with $\mathbf{X} \sim \mathbb{P}$, for all $g$ in $\mathcal{G}(k) = \{g : \mathbb{R}^d \to \mathbb{R}^d : \sum_{i=1}^d \|g_i\|_{\mathcal{H}(k)}^2 \leq 1\}$. The probability measure $\mathbb{P}$ on $\mathbb{R}^d$ is assumed to admit a continuously differentiable Lebesgue density $p \in \mathcal{C}^1(\mathbb{R}^d)$, such that $\mathbb{E}[\|\nabla \log p(\mathbf{X})\|_2^2] < \infty$. The Stein discrepancy is then defined as $\mathrm{SD}(\mathbb{P}, \mathbb{Q}) = \sup_{g \in \mathcal{G}(k)} |\mathbb{E}[(\mathcal{T}_p g)(\mathbf{Z})]|$, where $\mathbf{Z} \sim \mathbb{Q}$. If the Stein operator $\mathcal{T}_p$ is chosen as the Langevin operator $(\mathcal{T}_p g)(\mathbf{x}) = \langle g(\mathbf{x}), \nabla \log p(\mathbf{x}) \rangle + \langle \nabla, g(\mathbf{x}) \rangle$, then Stein's discrepancy has a closed-form expression known as kernelized Stein discrepancy [Chwialkowski et al., 2016, Liu et al., 2016], $\mathrm{KSD}^2(\mathbb{P}, \mathbb{Q}) = \mathbb{E}[k_p(\mathbf{Z}, \mathbf{Z}')]$, where $\mathbf{Z} \sim \mathbb{Q}, \mathbf{Z}' \sim \mathbb{Q}$, and $k_p$ denotes the Langevin Stein kernel defined from the score function $s_p(\mathbf{x}) = \nabla \log p(\mathbf{x})$ for $\mathbf{x}, \mathbf{x}' \in \mathbb{R}^d$, as

$$k_p(\mathbf{x}, \mathbf{x}') = \langle \nabla_{\mathbf{x}}, \nabla_{\mathbf{x}'} k(\mathbf{x}, \mathbf{x}') \rangle + \langle s_p(\mathbf{x}), \nabla_{\mathbf{x}'} k(\mathbf{x}, \mathbf{x}') \rangle$$
$$+ \langle s_p(\mathbf{x}'), \nabla_{\mathbf{x}} k(\mathbf{x}, \mathbf{x}') \rangle + \langle s_p(\mathbf{x}), s_p(\mathbf{x}') \rangle k(\mathbf{x}, \mathbf{x}'). \tag{2}$$

The main advantage of the KSD is that it only requires the knowledge of the score function, and does not involve any integration with respect to $\mathbb{P}$. Gorham and Mackey [2017] also established

convergence guaranties when the kernel $k$ is chosen as the inverse multi-quadratic (IMQ) kernel function $k(\mathbf{x}, \mathbf{x}') = (c + \|\mathbf{x} - \mathbf{x}'\|_\Gamma^2)^{-\beta}$ with $c > 0$, $\beta \in (0, 1)$, the positive definite matrix $\Gamma$ is the identity matrix, and the density $p$ is distantly dissipative as defined below. Log-concave distributions outside of a compact set are a typical example of such probability densities.

**Definition 1.1** (Distant dissipativity Gorham and Mackey [2017]). The density $p \in \mathcal{C}^1(\mathbb{R}^d)$ is distantly dissipative if $\liminf\limits_{r \to \infty} \kappa(r) > 0$, where $\kappa(r) = \inf \left\{ -2 \frac{\langle s_p(\mathbf{x}) - s_p(\mathbf{y}), \mathbf{x} - \mathbf{y} \rangle}{\|\mathbf{x} - \mathbf{y}\|_2^2} : \|\mathbf{x} - \mathbf{y}\|_2 = r \right\}$.

**Stein thinning algorithm.** Let $\mathbb{P}$ be a target probability measure that admits density $p$, and let $\{\mathbf{x}_i\}_{i=1}^n \subset \mathbb{R}^d$ be a MCMC output. The Stein thinning algorithm [Riabiz et al., 2022] selects $m \le n$ particles $\mathbf{x}_{\pi_1}, \ldots, \mathbf{x}_{\pi_m}$ by greedily minimizing the kernelized Stein discrepancy. Given $t - 1 < m$ particles $\mathbf{x}_{\pi_1}, \ldots, \mathbf{x}_{\pi_{t-1}}$, the $t$-th particle is defined as

$$\pi_t \in \operatorname*{argmin}_{i \in \{1, \ldots, n\}} k_p(\mathbf{x}_i, \mathbf{x}_i) + 2 \sum_{j=1}^{t-1} k_p(\mathbf{x}_{\pi_j}, \mathbf{x}_i),$$

where the KSD of an empirical distribution has been used to simplify the objective function. The kernel function $k$ is usually chosen as the IMQ kernel function, defined above, for both its good theoretical properties and empirical efficiency. Indeed, several articles [Chen et al., 2018, Riabiz et al., 2022] have led extensive experiments to show the better practical performance of the IMQ kernel over other choices. Also notice that the bandwidth parameter $\ell$ is quite influential on the algorithm performance, but happens to be very difficult to tune, as highlighted by Chopin and Ducrocq [2021]. Indeed, since the normalization constant of the target distribution is unknown, no additional metric is available to assess the precise performance of the thinning procedure when $\ell$ varies. Furthermore, the sample quality output by Stein thinning varies in an erratic fashion with respect to $\ell$, making the design of heuristic procedures for the choice of $\ell$ notoriously difficult. Following the literature recommendations [Riabiz et al., 2022], we use the median heuristic to set $\ell$ in our experiments, and refer to Garreau et al. [2017] for an extensive analysis of this approach for kernel methods.

## 2 Analysis of KSD Pathologies

Although kernelized Stein discrepancy is a highly promising approach to thin MCMC outputs, several empirical studies have highlighted that KSD-based algorithms may suffer from strong pathologies in simple experiments [Wenliang and Kanagawa, 2020, Korba et al., 2021, Riabiz et al., 2022, Liu et al., 2023]. The most established KSD pathology is that Stein thinning ignores the weights of distant modes of the target distribution, leading to the selection of samples of poor quality by Stein thinning. This problem, called Pathology I throughout the article, is analyzed in Subsection 2.1. Additionally, Korba et al. [2021] also notice that KSD thinning may result in samples concentrated in regions of low probability of $p$. As opposed to Pathology I, the mechanism leading to this problematic behavior is not well understood in the literature, to our best knowledge. Subsection 2.2 is thus dedicated to the theoretical characterization and illustration of Pathology II. Throughout the article, we illustrate KSD thinning using the running example of a Gaussian mixture, defined in Example 1 below, where initial particles are directly sampled from $p$ to better highlight pathologies. We will come back to the thinning of MCMC outputs in detail in Section 4.

**Example 1.** Let the density $p$ be a Gaussian mixture model of two components, respectively centered in $(-\mu, \mathbf{0}_{d-1})$ and $(\mu, \mathbf{0}_{d-1})$, of weights $w$ and $1 - w$, and of variance $\sigma^2 \mathbf{I_d}$. The initial particles $\{\mathbf{x}_i\}_{i=1}^n$ are drawn from $p$. The KSD thinning algorithm selects $m < n$ points to approximate $p$.

### 2.1 Pathology I: mode proportion blindness

We first focus on Pathology I, which states that Stein thinning is blind to the relative weights of multiple distant modes of a target distribution. Indeed, Wenliang and Kanagawa [2020] show that the score $s_p$ is insensitive to distant mode weights. Consequently, the KSD distance is unable to properly identify samples with different weights than those of the target, in finite sample settings, as long as samples are accurately distributed within each mode. To be more specific, we illustrate this pathology with our Example 1 of a Gaussian mixture in dimension 2. We set $\mu = 3$ and $\sigma = 1$ to enforce the two modes to be well separated, and take an unbalanced proportion $w = 0.2$ for the left mode, and $1 - w = 0.8$ for the right mode. We generate $n = 3000$ observations and select $m = 300$ particles with Stein thinning. Clearly, the red selected sample displayed in Figure 1 has wrong proportions, with about half of the

particles in each mode, instead of the expected $20 - 80\%$, reflected by the initial black particles sampled from $p$. More precisely, over 100 repetitions of the Stein thinning algorithm, we obtain an average proportion of $0.53$ particles in the left mode, with a standard deviation of $0.08$ across the 100 runs.

Although Wenliang and Kanagawa [2020] clearly show that the KSD distance is insensitive to the mode weights in the specific case of Gaussian mixtures, the mechanism leading to the selection of about half of the particles in each mode by Stein thinning, as in Example 1, remains unexplained in the literature, to our best knowledge. Therefore, we conduct a theoretical analysis in the general case of any mixture distribution with two distant modes, stated in Assumption 2.1 below. For the sake of clarity, we only study the case of a number of modes of two, without loss of generality. Importantly, notice that a finite sample drawn from a distribution with distant modes, takes the form of clusters of particles around each mode, as illustrated in Figure 1. Then, Stein thinning selects particles among these clusters to approximate $p$, and these particles define an empirical law of a density $q$ with a compact support around each mode. Wenliang and Kanagawa [2020] explain that the score $s_p$ is especially insensitive to the mode weights in these compact areas around modes, which is the root cause of the generation of samples with wrong proportions, as in Figure 1. Therefore, Assumption 2.1 below defines this observed setting, required to have Pathology I to occur, where density $q$ has compact supports around each distant mode. Additionally,

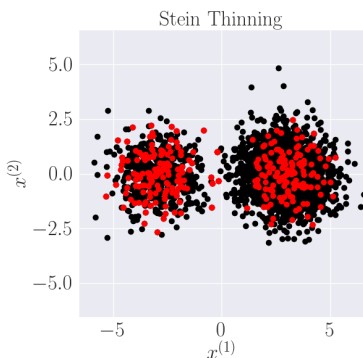

Figure 1: Illustration of Pathology I with the Gaussian mixture of Example 1 ($d = 2$, $\mu = 3$, $\sigma = 1$, $w = 0.2$, $n = 3000$, $m = 300$). Initial particles are in black, and the Stein thinning output is red.

we also need to formalize Assumption 2.2, which tells that the distributions of the two modes of the mixture $q$ have a close KSD distance with respect to the target $p$. In particular, this assumption can be easily verified when both $p$ and $q$ have symmetric mode distributions, since the KSD distance is insensitive to the weights of $p$.

**Assumption 2.1** (Distant bimodal mixture distributions). Let $p$ and $q$ be two mixture distributions in $\mathbb{R}^d$, made of two modes centered in $(-\mu, \mathbf{0}_{d-1})$ and $(\mu, \mathbf{0}_{d-1})$, with $\mu > 0$. The distribution of each mode of $p \in \mathcal{C}^1(\mathbb{R}^d)$ has $\mathbb{R}^d$ as support, whereas each mode distribution of $q$ have a compact support, included in a ball of radius $r > 0$, with $r < \mu$. The left mode of $p$ has weight $w_p \neq 1/2$, and the right mode has weight $1 - w_p$. Similarly, $w$ and $1 - w$ are the mode weights of $q$. Let $\mathbb{Q}_L$ and $\mathbb{Q}_R$ be the probability measures that respectively admit the density of the left and right modes of $q$, and $\mathbb{P}$ and $\mathbb{Q}_w$ be also the probability laws for $p$ and $q$.

**Assumption 2.2.** For distant bimodal mixture distributions $q$ and $p$ satisfying Assumption 2.1, and for $\eta \in (0, 1)$, we have $\left| \text{KSD}^2(\mathbb{P}, \mathbb{Q}_L)/\text{KSD}^2(\mathbb{P}, \mathbb{Q}_R) - 1 \right| < \eta$.

**Theorem 2.3.** *Let $k_p$ be the Stein kernel associated with the radial kernel $k(\mathbf{x}, \mathbf{x}') = \phi(\|\mathbf{x} - \mathbf{x}'\|_2/\ell)$, where $\mathbf{x}, \mathbf{x}' \in \mathbb{R}^d$, $\ell > 0$, and $\phi \in \mathcal{C}^2(\mathbb{R})$, such that $\phi(z) \to 0$, $\phi'(z) \to 0$, and $\phi''(z) \to 0$ for $z \to \infty$. Let $p$ and $q$ be two bimodal mixture distributions satisfying Assumptions 2.1 and 2.2, for any $\eta \in (0, 1)$. We define $w^\star$ as the optimal mixture weight of $q$ with respect to the KSD distance, i.e., $w^\star = \operatorname*{argmin}_{w \in [0,1]} \text{KSD}(\mathbb{P}, \mathbb{Q}_w)$. Then, for $\mu$ large enough, we have $\left| w^\star - \frac{1}{2} \right| < \frac{\eta}{2(1-\eta)}$.*

Theorem 2.3, proved in Appendix B, states that the weight $w^\star$ of the optimal mixture $q$, which minimizes the KSD distance to the target $p$, is close to $1/2$ regardless of the true target weight $w_p$, whenever the distributions of the two modes of the mixture $q$ have a close KSD distance to $p$, and provided that the two modes are distant. In particular, this is the case in the experiment of Example 1 and Figure 1, where the two modes are symmetric and well separated. Additionally, a more specific empirical illustration of Theorem 2.3 can be found in Appendix A.1. In Section 3, we will propose strategies improving Stein thinning to recover samples with accurate mode proportions.

## 2.2 Pathology II: spurious minimum

The core of this section is dedicated to the theoretical characterization of Pathology II. We first need to introduce additional notations to formalize our analysis. We thus define $\mathcal{M}_{s_0}$, the region of the input space where the score norm is lower than the threshold $s_0 \geq 0$, formally $\mathcal{M}_{s_0} = \{ \mathbf{x} \in$

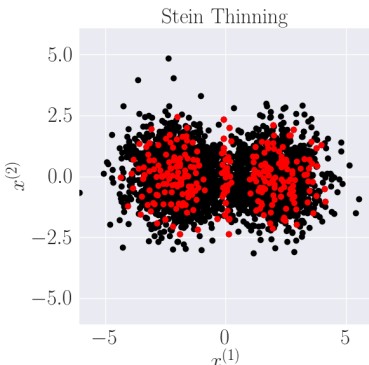 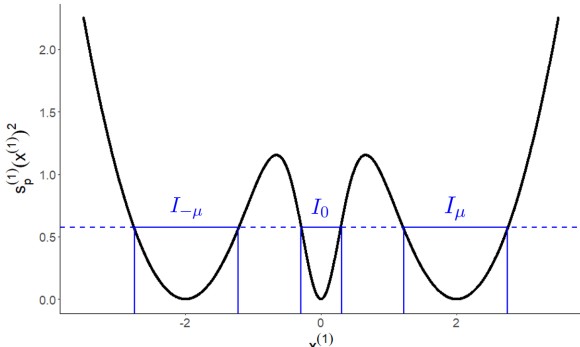

Figure 2: Illustration of Pathology II for the Gaussian mixture of Example 1 ($d = 2$, $\mu = 2$, $\sigma = 1$, $w = 0.5$, $n = 3000$, $m = 300$): many particles are selected around the line $x^{(1)} = 0$ (left panel), because of the squared first component of the score $s_p(\mathbf{x})$ along $x^{(1)}$ (for $x^{(2)} = 0$ in the right panel).

$\mathbb{R}^d : \|s_p(\mathbf{x})\|_2 \leq s_0\}$. We also introduce an independent and identically distributed (iid) sample $\mathbf{X}_1, \ldots, \mathbf{X}_m$ of $\mathbb{P}$, with $\mathbb{P}_m$ the associated empirical measure for a positive integer $m$, and $X^{(j)}$ the $j$-th component of $\mathbf{X}$. Then, Theorem 2.4 below shows that samples concentrated in regions of the input space where the norm of the score is low, have smaller KSD than samples drawn from the true target distribution $p$, for small sample sizes. Additionally, the score norm is low around stationary points of $p$, including local minimum and saddle points, as shown in Corollary 2.5 below. However, samples concentrated at local minimum of $p$ are bad approximations of the target distribution by definition. Therefore, pathological samples may be generated by Stein thinning, which minimizes the empirical KSD, and thus explains Pathology II observed by Korba et al. [2021], and shown in Figure 2. For the sake of clarity, we formalize our result for the IMQ kernel used in practice, and set $c = 1$ without loss of generality, since it is equivalent to tune $c$ or $\ell$ in the Stein thinning algorithm.

**Theorem 2.4** (KSD spurious minimum). *Let $k_p$ be the Stein kernel associated with the IMQ kernel with $\ell > 0$, $\beta \in (0,1)$, and $c = 1$. Let $\{\mathbf{x}_i\}_{i=1}^m \subset \mathcal{M}_{s_0} = \{\mathbf{x} \in \mathbb{R}^d : \|s_p(\mathbf{x})\|_2 \leq s_0\}$ be a fixed set of points of empirical measure $\mathbb{Q}_m = \frac{1}{m} \sum_{i=1}^m \delta(\mathbf{x}_i)$, with $s_0 \geq 0$ and $m \geq 2$. We have $\mathrm{KSD}^2\big(\mathbb{P}, \mathbb{Q}_m\big) < \mathbb{E}[\mathrm{KSD}^2\big(\mathbb{P}, \mathbb{P}_m\big)]$, if the score threshold $s_0$ and the sample size $m$ are small enough to satisfy $m < 1 + \big(\mathbb{E}[\|s_p(\mathbf{X})\|_2^2] - s_0^2\big)/(2\beta d/\ell^2 + 2\beta s_0/\ell + s_0^2)$.*

**Corollary 2.5** (Low KSD samples at density minimum). *Let $k_p$ be the Stein kernel associated with the IMQ kernel with $\ell > 0$, $\beta \in (0,1)$, and $c = 1$. Let $p$ be a density with at least one local minimum or saddle point. For $m \geq 2$, if $\{\mathbf{x}_i\}_{i=1}^m \subset \mathbb{R}^d$ is a set of points, all located at local minimum or saddle points of $p$, then we have $\mathrm{KSD}^2\big(\mathbb{P}, \mathbb{Q}_m\big) < \mathbb{E}[\mathrm{KSD}^2\big(\mathbb{P}, \mathbb{P}_m\big)]$, if $m < 1 + \frac{\ell^2}{2\beta d} \mathbb{E}[\|s_p(\mathbf{X})\|_2^2]$.*

The proofs of Theorem 2.4 and Corollary 2.5, reported in Appendix C, are built on the idea that the KSD of the empirical law of $\{\mathbf{x}_i\}_{i=1}^n$, has a bias of the form $\sum_{i=1}^m \|s_p(\mathbf{x}_i)\|_2^2/m^2$. Consequently, when $m$ is small, the bias has a strong influence on KSD estimates, which favor samples concentrated in regions of low score norm, as stationary points of $p$. This mechanism is illustrated in Figure 2 and Corollary 2.6 for Gaussian mixtures. In this case, Stein thinning aligns a large number of particles around the line of saddle points defined by $x^{(1)} = 0$, an area of low probability of the targeted mixture distribution, because of the variations of the score function, if the sample size $m$ is small enough. From another perspective, for any sample size $m$, it exists a Gaussian mixture with $\mu/\sigma$ large enough, such that Pathology II occurs. Therefore, Pathology II can appear for arbitrarily large samples $m$, depending on the target distribution properties.

**Corollary 2.6** (KSD spurious minimum for Gaussian mixtures). *Let $k_p$ be the Stein kernel associated with the IMQ kernel with $\ell > 0$, $\beta \in (0,1)$, and $c = 1$. Let the density $p$ be a Gaussian mixture model of two components with equal weights, respectively centered in $(-\mu, \mathbf{0}_{d-1})$ and $(\mu, \mathbf{0}_{d-1})$, of variance $\sigma^2 \mathbf{I_d}$, and let $\nu = \mu/\sigma$. If $\nu > 1$ and $0 \leq s_0 < \big[\nu\sqrt{\nu^2 - 1} - \ln(\nu + \sqrt{\nu^2 - 1})\big]/\mu$, then for any $\{\mathbf{x}_i\}_{i=1}^m \subset \mathcal{M}_{s_0}$ of empirical measure $\mathbb{Q}_m$, we have*

*(i) $\mathrm{KSD}^2\big(\mathbb{P}, \mathbb{Q}_m\big) < \mathbb{E}[\mathrm{KSD}^2\big(\mathbb{P}, \mathbb{P}_m\big)]$ if $m$ and $s_0$ satisfy $m < 1 + \frac{\mathbb{E}[\|s_p(\mathbf{X})\|_2^2] - s_0^2}{2\beta d/\ell^2 + 2\beta s_0/\ell + s_0^2}$,*

*(ii) there exists three disjoint intervals $I_{-\mu}, I_0, I_\mu \subset \mathbb{R}$, respectively centered around $-\mu$, $0$, and $\mu$, such that $x_1^{(1)}, \ldots, x_m^{(1)} \in I_{-\mu} \cup I_0 \cup I_\mu$.*

# 3 Regularized Stein Thinning

Stein thinning suffers from two main pathologies, analyzed in Section 2. In a word, Pathology I comes from the insensitivity of the score to the relative weights of distant modes, whereas Pathology II originates from the variations of the score norm, which do not differentiate local minimum from local maximum of the target distribution. We propose to regularize the KSD distance to fix these two problems, using terms that are highly sensitive to the type of stationary point and the relative weights of modes. The proposed algorithm is first introduced in Subsection 3.1, then theoretical properties are discussed in Subsection 3.2, and finally the good empirical performance will be shown in Section 4.

## 3.1 Algorithm

**Entropic regularization.** In order to compensate the blindness of the KSD to mode proportions in multimodal distributions, we introduce the following entropic regularized KSD, denoted by $\text{KSD}_\lambda$, and defined as $\text{KSD}_\lambda^2(\mathbb{P}, \mathbb{Q}) = \mathbb{E}[k_p(\mathbf{Z}, \mathbf{Z}')] - \lambda \mathbb{E}[\log(p(\mathbf{Z}))]$, where $\mathbf{Z}$ and $\mathbf{Z}'$ have probability law $\mathbb{Q}$, and $\mathbb{P}$ admits the density $p$. In our Bayesian setting, $\mathbb{E}[\log(p(\mathbf{Z}))]$ is known up to an additive constant since the normalization factor of $p$ is not tractable. However, it is possible to use $\text{KSD}_\lambda^2(\mathbb{P}, \mathbb{Q})$ as the objective function of the Stein thinning algorithm, as the greedy selection of particles to optimize this quantity does not rely on the unknown additive constant. The main idea of this entropic regularization is that $-\log(p(\mathbf{x}))$ takes higher values in modes of smaller probability, and therefore provides the relative mode weight information, which is missing in the KSD distance. More precisely, modes with smaller weights take smaller density values, and are therefore more penalized than modes of higher weights. Therefore, with such entropic penalization, regularized Stein thinning tends to select particles in modes of higher weights more frequently than in modes of smaller weights, and we recover appropriate proportions.

**Laplacian correction.** Chen et al. [2018] and Riabiz et al. [2022] have noticed that the term $k_p(\mathbf{x}_i, \mathbf{x}_i)$, which naturally appears in the empirical kernelized Stein discrepancy with the Langevin operator, can be interpreted as a regularization term. For example, Stein thinning does not select particles in the burn-in period of an MCMC output thanks to this regularization. However, this term $k_p(\mathbf{x}_i, \mathbf{x}_i)$ is also responsible for Pathology II, of samples concentrated in stationary points of $p$, as shown in Theorem 2.4. Therefore, we add a second regularization term to compensate the weaknesses of $k_p(\mathbf{x}_i, \mathbf{x}_i)$, by penalizing particles located at local minimum and saddle points of the density $p$. Such points are located in areas of convexity of the target distribution, which can thus be detected with the positive values of the Laplacian of the density. Therefore, using the truncated Laplacian operator $\Delta^+ f(\mathbf{x}) = \sum_{j=1}^d \left( \partial^2 f(\mathbf{x})/\partial x^{(j)2} \right)^+$ for a function $f \in \mathcal{C}^2(\mathbb{R}^d)$, we propose the L-KSD estimate with a Laplacian correction for densities $p \in \mathcal{C}^2(\mathbb{R}^d)$, defined by

$$\text{L-KSD}^2(\mathbb{P}, \mathbb{Q}_m) = \frac{1}{m^2} \sum_{i \neq j}^m k_p(\mathbf{x}_i, \mathbf{x}_j) + \frac{1}{m^2} \sum_{i=1}^m \left[ k_p(\mathbf{x}_i, \mathbf{x}_i) + \Delta^+ \log(p(\mathbf{x}_i)) \right].$$

**Regularized Stein thinning.** Overall, we obtain the following estimate for the entropic regularized KSD with Laplacian correction $\text{L-KSD}_\lambda^2(\mathbb{P}, \mathbb{Q}_m) = \text{L-KSD}^2(\mathbb{P}, \mathbb{Q}_m) - \frac{\lambda}{m} \sum_{i=1}^m \log(p(\mathbf{x}_i))$. Then, at each iteration $t \in \{1, \ldots, m\}$, the regularized Stein thinning selects the particle index $\pi_t \in \{1, \ldots, n\}$ to greedily minimize

$$k_p(\mathbf{x}_{\pi_t}, \mathbf{x}_{\pi_t}) + \Delta^+ \log(p(\mathbf{x}_{\pi_t})) - \lambda t \log(p(\mathbf{x}_{\pi_t})) + 2 \sum_{j=1}^{t-1} k_p(\mathbf{x}_{\pi_j}, \mathbf{x}_{\pi_t}).$$

Finally, Figure 3 illustrates the performance of regularized Stein thinning to fix the two pathologies analyzed in Section 2, in the case of Example 1 with Gaussian mixtures. Indeed, the top panel of Figure 3 shows that the majority of particles are selected in the right mode, as expected from the target distribution with $w = 0.2$. More precisely, an average proportion of $0.11$ of the particles are located in the left mode over $100$ repetitions of the procedure ($0.89$ in the right mode), with a standard deviation of $0.03$. For the value choice of $\lambda$, we refer to the next subsection and the experimental Section 4. On the bottom panel of Figure 3, we observe that no particle is now selected on the line $x^{(1)} = 0$, as expected from the target Gaussian mixture distribution.

**Remark 3.1.** The truncated Laplacian operator is simply given by the trace of the Hessian matrix, where negative components are set to $0$. It follows that the computational cost the regularized algorithm is similar to the original Stein thinning.

**Remark 3.2.** The Laplacian correction of $k_p$ introduces second-order derivatives of $p$ in the Stein discrepancy, and therefore enables to differentiate local minimum and saddle points of the density $p$ from its local maximum. A natural approach to introduce second-order derivatives of $p$ in KSD estimates, is to define the Stein discrepancy using second-order operators. A Laplacian Stein operator [Oates et al., 2017] is derived in Appendix G, but experiments show that this strategy is not efficient to fix Pathologies I & II.

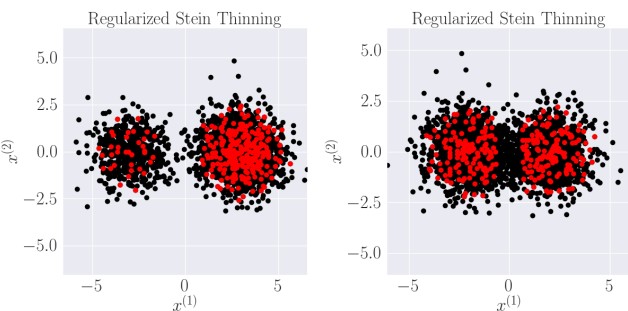

Figure 3: Pathologies fixed by the regularized Stein thinning.

## 3.2 Theoretical properties

This subsection is dedicated to the theoretical analysis of regularized Stein thinning. First, we show that the proposed algorithm now enjoys good properties regarding Pathologies I and II, and thus mitigates the identified problems of the original Stein thinning. Secondly, we extend the convergence analysis of Riabiz et al. [2022] for the post-treatment of MCMC output, to show the convergence of the empirical law output by regularized Stein thinning towards the target probability measure.

**Entropic regularization.** In the previous section, Theorem 2.3 highlights how Pathology I of mode proportion blindness originates from the score insensitivity to mode weights. On the other hand, the entropic regularization is directly built on the target density, and therefore strongly depends on the mode weights. In the same setting of Assumption 2.1, required for Pathology I to occur with the original algorithm, the following Theorem 3.3 shows that the entropic regularized KSD is minimized for the appropriate target weight, with the suitable regularization strength $\lambda$. Notice that Theorem 3.3, proved in Appendix D, is valid if $\mathbb{E}[\log(p(\mathbf{Z}_L))] \neq \mathbb{E}[\log(p(\mathbf{Z}_R))]$ with $\mathbf{Z}_L \sim \mathbb{Q}_L$ and $\mathbf{Z}_R \sim \mathbb{Q}_R$, otherwise the impact of the entropic regularization on $w_\lambda^\star$ vanishes. However, as $w_p \neq 1/2$ is required in Assumption 2.1 for Pathology I to occur, $p$ is asymmetric, and $\mathbb{E}[\log(p(\mathbf{Z}_L))] = \mathbb{E}[\log(p(\mathbf{Z}_R))]$ is only possible in very specific cases. Theorem 3.3 clearly shows that the regularized entropic KSD is sensitive to the weights of distant modes. Efficient strategies to choose the regularization strength will be first discussed in the asymptotic analysis below, and then in the experiments of the next section.

**Theorem 3.3.** *Let $k_p$ be the Stein kernel associated with the radial kernel $k(\mathbf{x}, \mathbf{x}') = \phi(\|\mathbf{x}-\mathbf{x}'\|_2/\ell)$, where $\mathbf{x}, \mathbf{x}' \in \mathbb{R}^d$, $\ell > 0$, and $\phi \in \mathcal{C}^2(\mathbb{R})$. Let $p$ and $q$ be two bimodal mixture distributions satisfying Assumption 2.1. We define $w_\lambda^\star$ as the optimal mixture weight of $q$ with respect to the entropic regularized KSD distance, i.e., $w_\lambda^\star = \underset{w \in [0,1]}{\operatorname{argmin}} \, \mathrm{KSD}_\lambda(\mathbb{P}, \mathbb{Q}_w)$. If $\mathbb{E}[\log(p(\mathbf{Z}_L))] \neq \mathbb{E}[\log(p(\mathbf{Z}_R))]$ where $\mathbf{Z}_L \sim \mathbb{Q}_L$ and $\mathbf{Z}_R \sim \mathbb{Q}_R$, it exists $\lambda \in \mathbb{R}$ such that $w_\lambda^\star = w_p$.*

**Laplacian correction.** First, we stress that the L-KSD is a strongly consistent estimate of the KSD distance, where the proof follows from the law of large numbers. Therefore, the Laplacian correction introduced in the L-KSD estimate does not undermine the good asymptotic properties of the KSD distance. Secondly, the following theorem shows that samples concentrated in local minimum or saddle points of the target distribution and of low density values, are well identified by the L-KSD as samples of worse quality than those truly sampled from the target. Consequently, the Laplacian correction fixes Pathology II, previously formalized in Theorem 2.4.

**Theorem 3.4.** *Let $k_p$ be the Stein kernel associated with the IMQ kernel with $\ell > 0$, $\beta \in (0, 1)$, and $c = 1$. For $m \geq 2$, let $\{\mathbf{x}_i\}_{i=1}^m \subset \mathbb{R}^d$ be a set of points located at $\mathbf{x}_0$, a local minimum or saddle point of $p$, and of empirical measure $\mathbb{Q}_m$. Then, we have $\mathrm{L\text{-}KSD}^2(\mathbb{P}, \mathbb{Q}_m) > \mathbb{E}[\mathrm{L\text{-}KSD}^2(\mathbb{P}, \mathbb{P}_m)]$, if the density at $\mathbf{x}_0$ satisfies $p(\mathbf{x}_0) < \Delta^+ p(\mathbf{x}_0)/(\mathbb{E}[\|s_p(\mathbf{X})\|_2^2] + \mathbb{E}[\Delta^+ \log p(\mathbf{X})])$.*

**Convergence of regularized Stein thinning.** While regularized Stein thinning fixes finite sample size pathologies, the asymptotic properties of Stein thinning are also preserved. Indeed, if the initial set of particles is drawn from a different distribution than the target using a Markov chain Monte Carlo, Theorem 3.6 states that the empirical measure of the sample obtained with regularized Stein Thinning, converges towards the target measure $\mathbb{P}$, and thus extends the results of Riabiz et al. [2022]. Notice that the weak convergence of a sequence of probability measure is denoted by $\Rightarrow$, and that distantly dissipative distributions are defined in Definition 1.1. The required assumption below, essentially states mild integrability conditions, and enforces that the MCMC output is not too far from a sample drawn from $p$—see Appendix F for additional details.

**Assumption 3.5.** Let $\mathbb{Q}$ be a probability distribution on $\mathbb{R}^d$, such that $\mathbb{P}$ is absolutely continuous with respect to $\mathbb{Q}$. Let $\{\mathbf{Z}_i\}_{i\in\mathbb{N}} \subset \mathbb{R}^d$ be a $\mathbb{Q}$-invariant, time-homogeneous Markov chain, generated using a $V$-uniformly ergodic transition kernel, such that $V(\mathbf{x}) \geq \frac{d\mathbb{P}}{d\mathbb{Q}}\sqrt{2\beta d/\ell^2 + \|s_p(\mathbf{x})\|_2^2}$. Suppose that, for some $\gamma > 0$, $\sup_{i\in\mathbb{N}} \mathbb{E}[e^{\gamma|\log(p(\mathbf{Z}_i))|}] < \infty$, $\sup_{i\in\mathbb{N}} \mathbb{E}[e^{\gamma\Delta^+ \log p(\mathbf{Z}_i)}] < \infty$,

$$\sup_{i\in\mathbb{N}} \mathbb{E}\big[e^{\gamma\max(1,\frac{d\mathbb{P}}{d\mathbb{Q}}(\mathbf{Z}_i)^2)(\frac{2\beta d}{\ell^2}+\|s_p(\mathbf{Z}_i)\|_2^2)}\big] < \infty, \quad \sup_{i\in\mathbb{N}} \mathbb{E}\Big[\frac{d\mathbb{P}}{d\mathbb{Q}}(\mathbf{Z}_i)\sqrt{\frac{2\beta d}{\ell^2}+\|s_p(\mathbf{Z}_i)\|_2^2}V(\mathbf{Z}_i)\Big] < \infty.$$

**Theorem 3.6.** *Let $\mathbb{P}$ be a distantly dissipative probability measure, that admits the density $p \in \mathcal{C}^2(\mathbb{R}^d)$, $k_p$ be the Stein kernel associated with the IMQ kernel where $\ell, c > 0, \beta \in (0,1)$. Let $\{\mathbf{Z}_i\}_{i\in\mathbb{N}} \subset \mathbb{R}^d$ be a Markov chain satisfying Assumption 3.5, $\pi$ be the index sequence of length $m_n$ generated by regularized Stein thinning, and $\mathbb{Q}_{m_n}$ be the empirical measure of $\{\mathbf{Z}_{\pi_i}\}_{i=1}^{m_n}$. If $\log(n)^\alpha < m_n < n$, with any $\alpha > 1$, and $\lambda_{m_n} = o(\log(m_n)/m_n)$, then we have almost surely $\mathbb{Q}_{m_n} \underset{n\to\infty}{\Longrightarrow} \mathbb{P}$.*

Theorem 3.6, proved in Appendix F, provides us with interesting insights about the entropic regularization strength $\lambda$. We already know that $\lambda$ should be chosen with a rate at least as fast as $O(1/m)$, to avoid the introduction of a higher a bias in the L-KSD than the original KSD. Indeed, for a sample drawn from the true target distribution $p$, this bias $\mathbb{E}[\text{L-KSD}^2(\mathbb{P}, \mathbb{P}_m)]$ takes the form $\mathbb{E}[k_p(\mathbf{X}, \mathbf{X})]/m + \mathbb{E}[\Delta^+ \log(p(\mathbf{X}))]/m - \lambda\mathbb{E}[\log(p(\mathbf{X}))]$. Therefore, for slower rates of $\lambda$ than $O(1/m)$, trivial samples concentrated at a local maximum of the target $p$, can have smaller L-KSD than samples drawn from $p$. Then, Theorem 3.6 states that, for our ultimate application of MCMC post-processing, the Stein thinning sample distribution converges towards the target for such $\lambda$ rates of $O(1/m)$ or faster. In practice, in our Bayesian setting, it is not possible to fine tune this parameter $\lambda$ because no metric is available to assess the Stein thinning quality for various values of $\lambda$, as already mentioned in the case of the bandwidth parameter $\ell$. In addition, we cannot theoretically determine which exact range of values of $\lambda$ leads to good thinned samples in a finite sample regime. However, we will see in the experiments of the following section that both slower and faster $\lambda$ rates than $O(1/m)$ lead to samples of degraded quality. Therefore, we set $\lambda = 1/m$ in the regularized Stein thinning, to ensure good empirical performance and the algorithm convergence.

## 4 Empirical Assessment

This section shows how regularized Stein thinning outperforms the original algorithm through three batches of experiments: mixtures of standard distributions using exact or MCMC sampling, and Bayesian logistic regression on real datasets. For the experiments considered in Sections 4.2 and 4.3, two Metropolis-Hastings samplers are considered with the Metropolis-Adjusted Langevin Algorithm (MALA) and the No-U-Turn sampler (NUTS). We use the IMQ kernel with $\ell$ set with the median heuristic, $\beta = 1/2$, and $c = 1$, as recommended in Chen et al. [2018], Riabiz et al. [2022]. We also set the regularization parameter with the default value of $\lambda = 1/m$. Notice that additional experiments are provided in Appendix A, and that the code is available at https://gitlab.com/drti/kernax.

When the target distribution is known, the efficiency of the Stein thinning algorithms are assessed by computing the MMD distance (see Equation (1)) between a large sample drawn from the target distribution and the thinned samples. More specifically, we use the following closed-form expression of the MMD [Gretton et al., 2006] with $\mathbf{X}, \mathbf{X}' \sim \mathbb{P}$ and $\mathbf{Z}, \mathbf{Z}' \sim \mathbb{Q}$,

$$\text{MMD}_k^2(\mathbb{P}, \mathbb{Q}) = \mathbb{E}[k(\mathbf{X}, \mathbf{X}')] + \mathbb{E}[k(\mathbf{Z}, \mathbf{Z}')] - 2\mathbb{E}[k(\mathbf{X}, \mathbf{Z})], \tag{3}$$

where the kernel function $k$ is chosen as the distance-induced kernel studied by Sejdinovic et al. [2013] and given by $k(\mathbf{x}, \mathbf{x}') = \|\mathbf{x}\|_2 + \|\mathbf{x}'\|_2 - \|\mathbf{x} - \mathbf{x}'\|_2$, for $\mathbf{x}, \mathbf{x}' \in \mathbb{R}^d$. In this setting, the MMD reduces to the well known energy distance, as shown by Sejdinovic et al. [2013].

## 4.1 Gaussian mixtures with exact sampling

As a first batch of experiments, we build on Example 1 and consider more complicated two-dimensional Gaussian mixtures to further illustrate the correction of Pathologies I & II. The first Gaussian mixture is made of four modes located at $\boldsymbol{\mu}_1 = (-3, 3)$, $\boldsymbol{\mu}_2 = (-3, 3)$, $\boldsymbol{\mu}_3 = (3, 3)$, and $\boldsymbol{\mu}_4 = (3, -3)$, and with weights $w_1 = w_2 = 0.1$, and $w_3 = w_4 = 0.4$, respectively. The second mixture is taken from [Qiu and Wang, 2023]. It is made of 6 equally weighted Gaussian distributions centered at $\boldsymbol{\mu}_i = (3\cos(2\pi(i-1)/6), 3\sin(2\pi(i-1)/6))$, for $i = 1, \ldots, 6$. For both experiments, we rely on exact Monte Carlo sampling to generate $n = 3000$ observations, and apply Stein thinning and its regularized variant to select $m = 300$ particles. The observed samples and the selected particles are shown in Figure 4. The first example shows that vanilla Stein thinning does not capture the right proportions, while the regularized variant appropriately penalizes modes with lower weights. The second example illustrates Pathology II, which is corrected by the regularized Stein thinning.

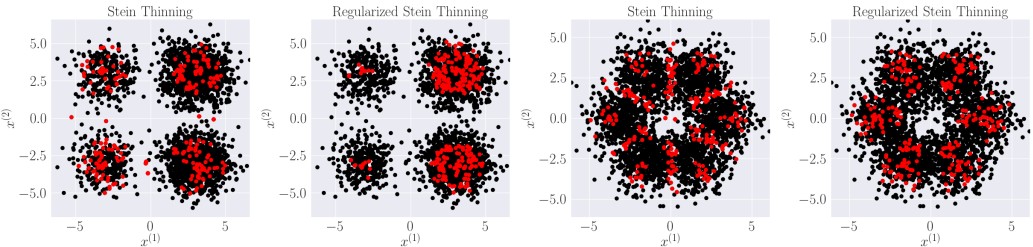

Figure 4: Gaussian mixtures with exact Monte Carlo sampling. Solutions (red dots) obtained by Stein thinning and its regularized variant.

## 4.2 Banana-shaped and Gaussian mixtures with MCMC sampling

We consider a mixture of two distant modes of $d$-dimensional banana-shaped distributions with t-tails and unbalanced weights [Haario et al., 1999, Pompe et al., 2020], illustrated in Figure 5, and precisely defined in Appendix A.2. We sample this target banana mixture with both MALA and NUTS using three different step sizes $\varepsilon$ and $10^5$ iterations. The generated samples are post-processed with the Stein thinning and regularized Stein thinning algorithms, and their performances are compared with the MMD between the post-processed samples and large samples drawn from the known target banana mixture. This experiment is run for various thinning sizes $m$ and dimensions $d$, with 20 repetitions to quantify uncertainties. The results obtained with the MALA sampler are shown in Figure 6: the regularized Stein thinning clearly generates samples of higher quality than the vanilla Stein thinning. Additionally, an example of post-processed MALA output is depicted in Figure 5, together with a heatmap of the Laplacian correction. On the left panel of Figure 5, we see that pathologies are especially strong in this experiment, with a large number of particles lying between the two modes in a region of low probability. On the right panels, we observe that regularized Stein thinning fix pathologies. Similar results were obtained with NUTS and are reported in Appendix A.2 for brevity. Next, we conduct the same experiments for a $d$-dimensional Gaussian mixture of four modes with different variances, as detailed in Appendix A.2. Again, Figure 6 shows the better

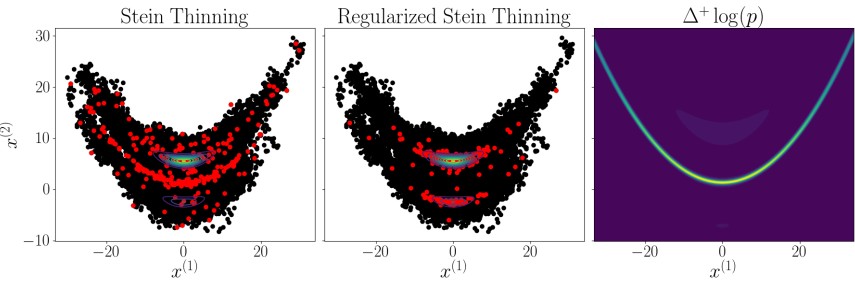

Figure 5: t-banana-shaped mixture ($d = 10$). From left to right: solutions obtained with standard and regularized Stein thinning with contour lines of $p$, and heatmap of the Laplacian correction.

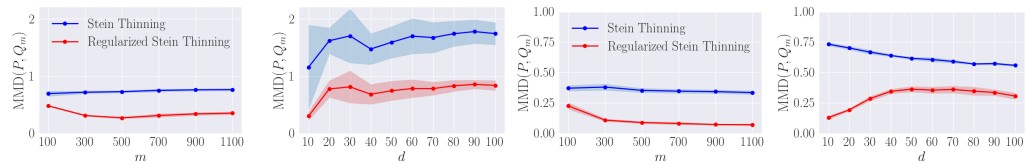

Figure 6: (MALA) Graphs of the MMD with respect to the thinning size $m$ (for $d = 2$) and with respect to $d$ (for $m = 300$). Left two panels: banana mixture. Right two panels: Gaussian mixture.

performance of regularized Stein thinning. Besides, we take advantage of this last experiment to explore other regularization rates than our default $\lambda = 1/m$. Figure 12 in Appendix A.2 shows that a slower rate of $\lambda = 1/\log(m)$, which violates the convergence assumptions of Theorem 3.6, has significantly worse performance than the original Stein thinning. On the other hand, with a faster rate than $1/m$ such as $1/m^2$, the effect of the entropic regularization disappears, and we recover similar results than the original Stein thinning. This supports that the default value of $\lambda = 1/m$ is an efficient heuristic, since slower and faster rates of $\lambda$ strongly degrade the algorithm performance.

### 4.3 Bayesian logistic regression

We now compare the two Stein thinning algorithms in the Bayesian logistic regression setting for binary classification, since such problem usually involves multimodal posterior—see, *e.g.*, Gershman et al. [2012], Liu and Wang [2016], Fong et al. [2019], Korba et al. [2021]. Given a dataset $\mathcal{D}_N = \{(\mathbf{X}_i, Y_i)\}_{i=1}^N$ made of $N$ pairs of features $\mathbf{X}_i \in \mathbb{R}^d$ and labels $Y_i \in \{0, 1\}$, the probability that $Y_i$ is of class 1 is given by $p(Y_i = 1|\mathbf{X}_i, \boldsymbol{\beta}, \beta_0) = 1/(1 + \exp(-\beta_0 - \boldsymbol{\beta}^T \mathbf{X}_i))$, for some parameters $\boldsymbol{\theta} = (\beta_0, \boldsymbol{\beta}) \in \mathbb{R}^{d+1}$. The prior distributions of the weight vector $\boldsymbol{\theta}$ is assumed to be Gaussian, $p(\beta^{(j)}|\gamma^{(j)}) = \mathcal{N}(\beta^{(j)}|0, 1/\gamma^{(j)})$, and a Gamma prior with parameters $(a, b)$ is chosen for the precision $\gamma^{(j)}$. Following [Fong et al., 2019], the hyperparameters are chosen as $a = b = 1$.

The posterior distribution of the weights $\boldsymbol{\theta}$ is sampled with both MALA and NUTS using 48 independent chains, of respectively $10^4$ and $10^5$ iterations, and four step sizes $\varepsilon$ are considered along with three thinning sizes $m$. Each MCMC sample is post-processed with the two Stein thinning algorithms. For a new input $\mathbf{x}^\star$, the resulting thinned samples are used to approximate the posterior predictive distribution $p(Y = 1|\mathbf{x}^\star, \mathcal{D}_N)$, defined by $\int p(Y = 1|\mathbf{x}^\star, \boldsymbol{\theta}) p(\boldsymbol{\theta}|\mathcal{D}_N) d\boldsymbol{\theta}$. The per-

Table 1: AUCs obtained with NUTS sampler for Stein Thinning (ST) and Regularized Stein Thinning (RST).

| Dataset | $m = 50$ | | $m = 300$ | |
|---|---|---|---|---|
| | ST | RST | ST | RST |
| Breast W. | 0.88 (0.02) | **0.96** (0.00) | 0.93 (0.01) | **0.96** (0.00) |
| Diabetes | **0.52** (0.01) | 0.50 (0.02) | 0.53 (0.02) | **0.57** (0.02) |
| Haberman | 0.51 (0.04) | **0.53** (0.02) | 0.53 (0.03) | **0.58** (0.02) |
| Liver | 0.53 (0.04) | **0.69** (0.01) | 0.61 (0.04) | **0.70** (0.01) |
| Sonar | 0.80 (0.02) | **0.81** (0.01) | 0.81 (0.01) | 0.81 (0.01) |

formance of Stein thinning algorithms are assessed using the standard AUC metric for classification problems, estimated with 10-fold cross-validation and 10 repetitions for uncertainties. Table 1 gathers the results for five public datasets from the UCI repository [Dua and Graff, 2017], and described in Appendix A.3, where the best AUC obtained for each algorithm over the four MCMC step sizes are reported. Clearly, regularized Stein thinning significantly improves the performance of Bayesian logistic regression.

## 5 Conclusion

Stein thinning has raised a high interest in recent years, as a powerful tool to post-process MCMC outputs, by the greedy minimization of the kernelized Stein discrepancy. Unfortunately, empirical studies have shown that KSD-based algorithms suffer from strong pathologies. We have conducted an in-depth theoretical analysis to identify the mechanisms at stake. From this understanding, we propose an improved Stein thinning algorithm relying on entropic regularization and Laplacian correction. This approach exhibits relevant theoretical properties regarding pathologies, as well as highly improved empirical performance. Finally, the analysis of these regularization terms for other types of KSD-based algorithms, such a KSD descent, seems a promising route for future work.

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

# Appendix

## A  Additional Experiments

### A.1  Illustration of Theorem 2.3

To better illustrate Theorem 2.3, we run an additional experiment, where $p$ is still defined as in Figure 7 from Example 2 recalled below, with unbalanced mode weights of $0.2$ and $0.8$. The density $q$ is distributed as $p$, but each mode is truncated outside a circle of two standard deviation radius, and $q$ has weight $w$. Next, for various values of $w \in [0.1, 0.9]$, we draw two samples of size $n = 3000$ from $p$ and $q$, and compute $\mathrm{KSD}(\mathbb{P}, \mathbb{Q}_w)$ (with 30 repetitions for each $w$ value). The result is displayed in Figure 8, and shows that the optimal weight is close to $1/2$, as predicted by Theorem 2.3, since $|\mathrm{KSD}^2(\mathbb{P}, \mathbb{Q}_L)/\mathrm{KSD}^2(\mathbb{P}, \mathbb{Q}_R) - 1|$ is estimated as $0.01$ in this case, implying that $|w^\star - 1/2| < 0.005$.

**Example 2.** Let the density $p$ be a Gaussian mixture model of two components, respectively centered in $(-\mu, \mathbf{0}_{d-1})$ and $(\mu, \mathbf{0}_{d-1})$, of weights $w$ and $1 - w$, and of variance $\sigma^2 \mathbf{I_d}$. The initial particles $\{\mathbf{x}_i\}_{i=1}^n$ are drawn from $p$. The KSD thinning algorithm selects $m < n$ points to approximate $p$.

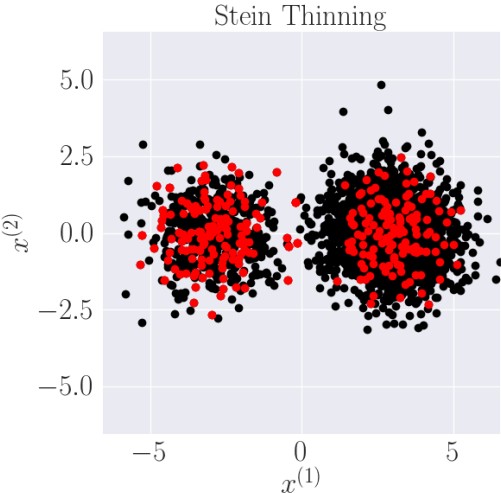

Figure 7: Illustration of Pathology I with the Gaussian mixture of Example 2 ($d = 2$, $\mu = 3$, $\sigma = 1$, $w = 0.2$, $n = 3000$, $m = 300$). Initial particles are in black, and the Stein thinning output is red.

### A.2  Gaussian and Banana-shaped Mixtures

This appendix gathers additional results and details for the Gaussian and banana-shaped mixtures, as well as the MMD distance used to evaluate thinning performance, and the regularization parameter $\lambda$.

**Gaussian mixture.**  The second batch of experiments in Section 4 considers a $d$-dimensional Gaussian mixture of four modes of equal weight, with $d \geq 2$, illustrated in Figure 9. The center of modes are chosen as $(-2, 0)$, $(2, 0)$, $(-3, 4)$, and $(3, 4)$, and null values for the higher dimension coordinates. The first two modes have an identity covariance matrix, while the remaining two modes have a diagonal covariance matrix with variance equal to $2$. The results for regularized Stein thinning and the original Stein thinning are provided in Figure 10 for MALA sampler, and in Figure 11 for NUTS sampler. In both figures, the three tested step size $\varepsilon$ are displayed, with a small impact on the resulting performance. Figures 9, 10, and 11 show the high performance improvement of regularized Stein thinning over the original algorithm.

**Regularization parameter $\lambda$.**  Figure 12 displays the MMD obtained with regularization parameters $\lambda$, set as $\lambda = 1/m^2$ and $\lambda = 1/\log(m)$. These results should be compared with the ones shown

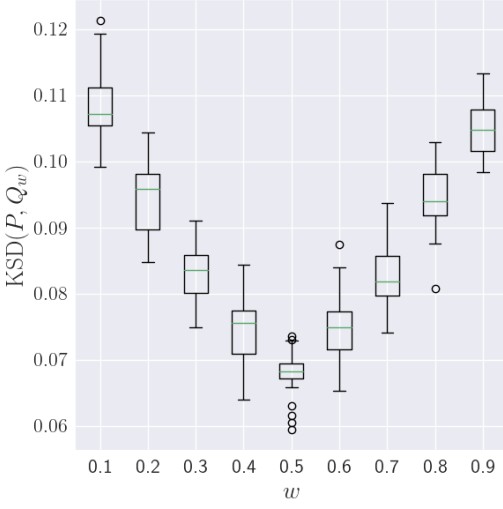

Figure 8: $KSD(\mathbb{P}, \mathbb{Q}_w)$ for $p$ as defined in Example 2 with $\mu = 3$, $\sigma = 1$, $w_p = 0.2$, and $q$ a truncation of $p$ and with weight $w$. The KSD is estimated with $n = 3000$ and 30 repetitions for each $w$ value.

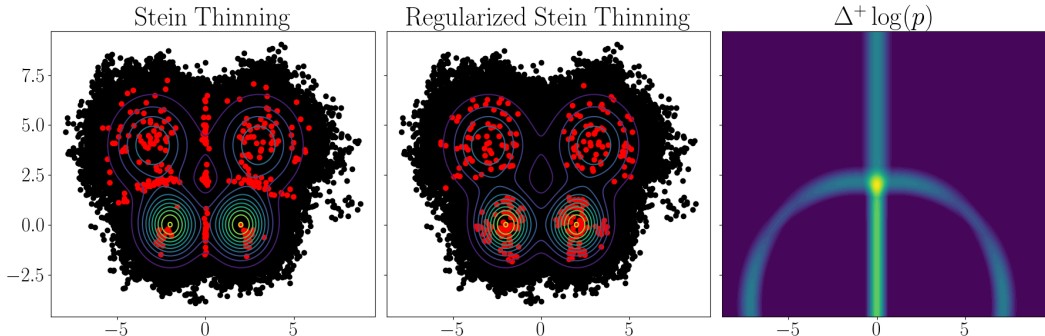

Figure 9: (Gaussian mixture with $d = 2$, MALA) First two panels: solutions obtained with Stein thinning and regularized Stein thinning with contour lines of the target distribution. Last panel: heatmap of the Laplacian correction $\Delta^+ \log(p)$.

in Figures 10 and 11, which were obtained with a regularization parameter $\lambda = 1/m$. These additional experiments show the importance of choosing the regularization parameter as $\lambda = O(1/m)$, as suggested by Theorem 3.6. Indeed, slower rates of $\lambda$ give poor quality samples, and faster rates than $\lambda = O(1/m)$ tend to remove the effect of the entropic regularization, and we then recover similar performance than the original Stein thinning. On the other hand, $\lambda = 1/m$ provides a high improvement over the standard thinning, as shown in Figures 10 and 11.

**Banana-shaped mixture with t-tails.** The first batch of experiments in Section 4 considers a banana-shaped mixture with t-tails, defined as follows. Let $\varphi : \mathbb{R}^d \to \mathbb{R}^d$ be the transformation defined by $\varphi_i(x) = x_i$ if $i \neq 2$, and $\varphi_2(x) = x_2 + bx_1^2 - 100b$. Let $\mathbf{Z}$ be a random variable that follows the multivariate t-Student distribution with degrees of freedom 7. Then, the random variable $\mathbf{X} = \varphi(\mathbf{Z}) + \boldsymbol{\mu}$ follows a t-banana-shaped distribution centered at $\boldsymbol{\mu}$. We consider a mixture of two t-banana-shaped distributions centered in $\mathbf{0}_d$ and $(0, 8, \mathbf{0}_{d-2})$, with weights $w_1 = 0.25$ and $w_2 = 0.75$, respectively, which is illustrated in Figure 13 for $d = 10$. The results for regularized Stein thinning and the original Stein thinning are provided in Figure 14 for MALA sampler, and in Figure 15 for NUTS sampler. In both figures, the three tested step size $\varepsilon$ are displayed, which confirms the higher performance of regularized Stein thinning. Figures 13, 14, and 15 show the high performance improvement of regularized Stein thinning over the original algorithm.

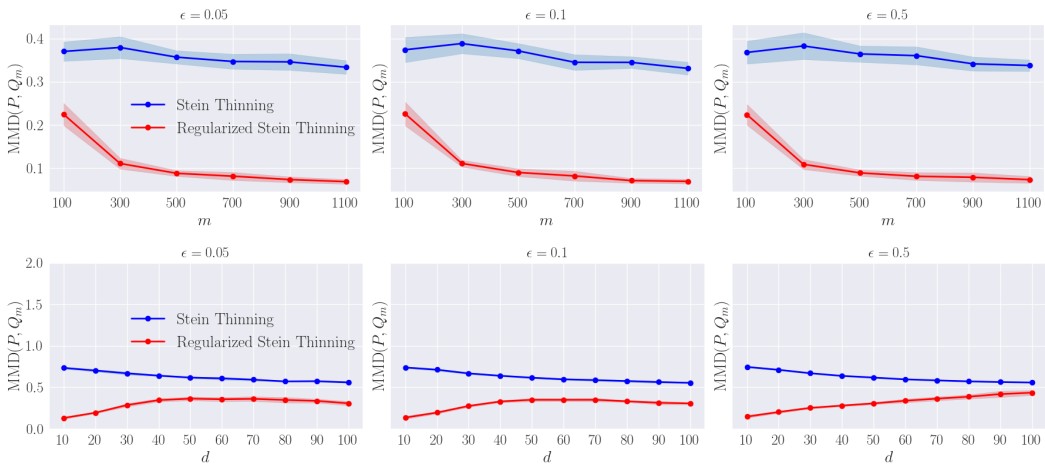

Figure 10: (Gaussian mixture, MALA) Graphs of the MMD distance with respect to the thinning size $m$ (with $d = 2$) and with respect to $d$ (with $m = 300$) for various step sizes $\varepsilon$.

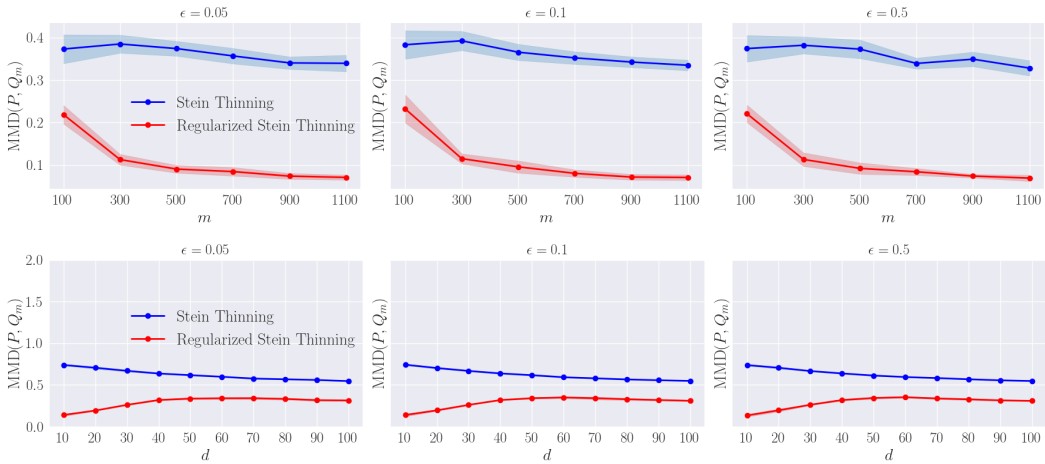

Figure 11: (Gaussian mixture, NUTS) Graphs of the MMD distance with respect to the thinning size $m$ (with $d = 2$) and with respect to $d$ (with $m = 300$) for various step sizes $\varepsilon$.

### A.3  Bayesian Logistic Regression

This appendix gathers additional results for Bayesian logistic regression. In particular, Table 2 provides a description of the tested datasets. Table 3 gives the resulting AUC, for $m = 50, 100, 300$, using NUTS or MALA sampler. We recall that only the best AUC over the four tested MCMC step size $\varepsilon$ is reported.

Recall that the Bayesian logistic regression defines the probability that $Y_i$ is of class 1 as $p(Y_i = 1|\mathbf{X}_i, \boldsymbol{\beta}, \beta_0) = 1/(1 + \exp(-\beta_0 - \boldsymbol{\beta}^T \mathbf{X}_i))$, for some vector of parameters $\boldsymbol{\theta} = (\beta_0, \boldsymbol{\beta}) \in \mathbb{R}^{d+1}$. The prior distributions of the weight vector $\boldsymbol{\theta}$ is assumed to be Gaussian, $p(\beta^{(j)}|\gamma^{(j)}) = \mathcal{N}(\beta^{(j)}|0, 1/\gamma^{(j)})$, and a Gamma prior with parameters $(a, b)$ is chosen for the precision $\gamma^{(j)}$. Upon marginalizing, it is found that $\beta^{(j)}$ is distributed as the non-standardized t-distribution Student-t$(2a, 0, b/a)$ [Bishop and Nasrabadi, 2006]. Following [Fong et al., 2019], the hyperparameters are chosen as $a = b = 1$.

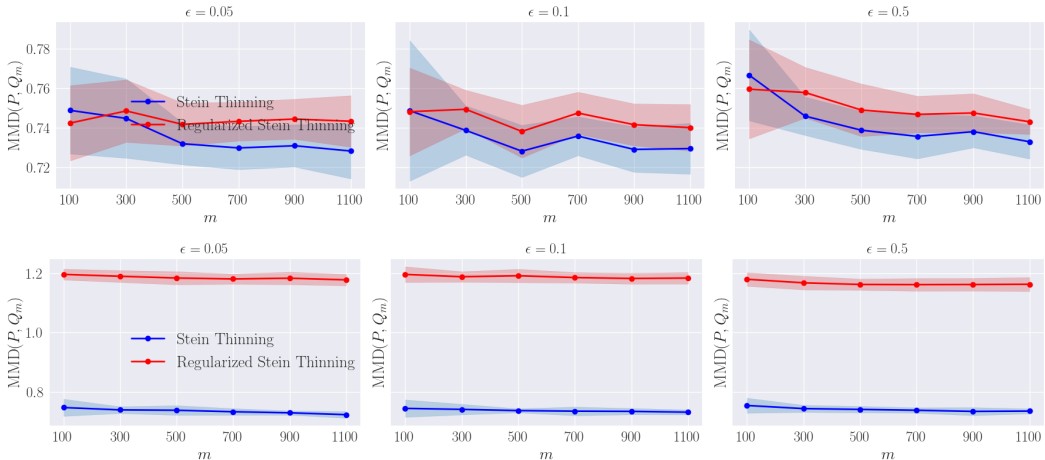

Figure 12: (Gaussian mixture, MALA) Graphs of the MMD distance with respect to the thinning size $m$ (with $d = 10$) for various step sizes $\varepsilon$. For the first row, we set $\lambda = 1/m^2$, and we observe that the effect of entropic regularization almost vanishes, since the performance is close to the original Stein thinning. For the second row, we set $\lambda = 1/\log(m)$, violating the convergence assumption, and resulting in bad thinned samples.

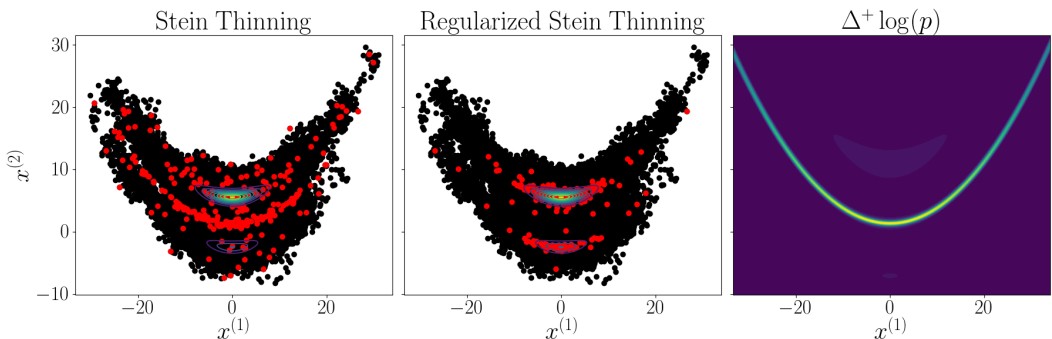

Figure 13: (t-banana-shaped mixture with $d = 10$, MALA) First two panels: solutions obtained with Stein thinning and regularized Stein thinning with contour lines of the target distribution. Last panel: heatmap of the Laplacian correction $\Delta^+ \log(p)$ for $x^{(3)} = \ldots = x^{(10)} = 0$.

Table 2: Description of UCI datasets

| Dataset | Sample size | Dimension |
|---|---|---|
| Breast Wisconsin | 569 | 30 |
| Diabetes | 768 | 8 |
| Haberman | 306 | 3 |
| Liver Disorders | 345 | 6 |
| Sonar | 208 | 60 |

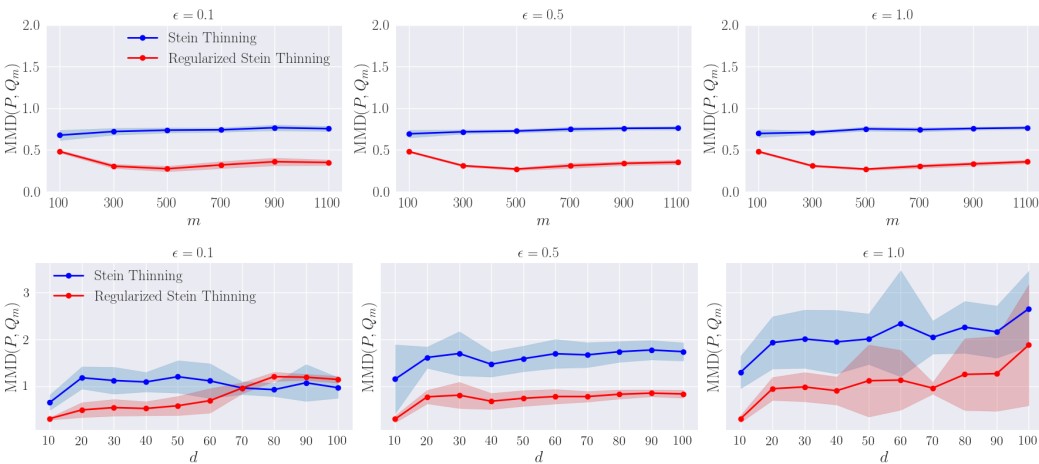

Figure 14: (Mixture of t-banana-shaped distributions, MALA) Graphs of the MMD distance with respect to the thinning size $m$ (with $d = 2$) and with respect to $d$ (with $m = 300$) for various step sizes $\varepsilon$.

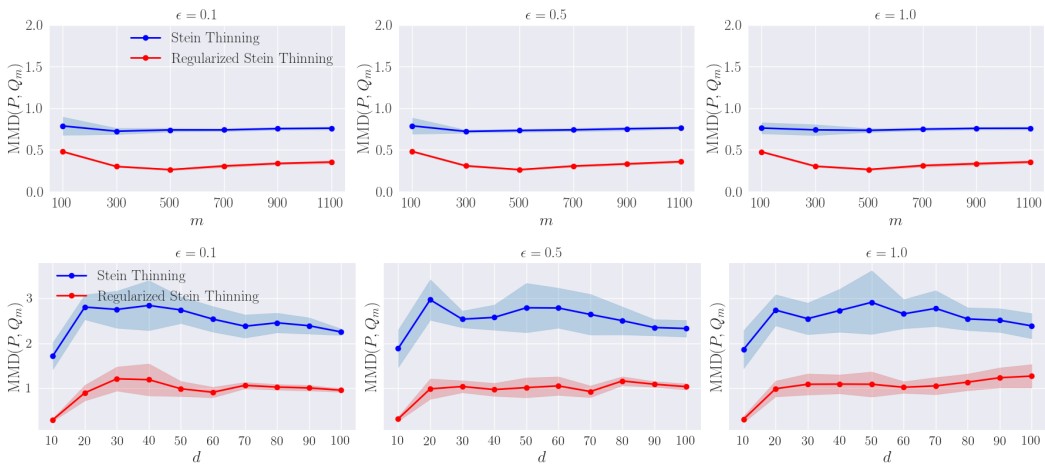

Figure 15: (Mixture of t-banana-shaped distributions, NUTS) Graphs of the MMD distance with respect to the thinning size $m$ (with $d = 2$) and with respect to $d$ (with $m = 300$) for various step sizes $\varepsilon$.

# B Proof of Theorem 2.3

**Assumption 2.1** (Distant bimodal mixture distributions). Let $p$ and $q$ be two mixture distributions in $\mathbb{R}^d$, made of two modes centered in $(-\mu, \mathbf{0}_{d-1})$ and $(\mu, \mathbf{0}_{d-1})$, with $\mu > 0$. The distribution of each mode of $p \in \mathcal{C}^1(\mathbb{R}^d)$ has $\mathbb{R}^d$ as support, whereas each mode distribution of $q$ have a compact support, included in a ball of radius $r > 0$, with $r < \mu$. The left mode of $p$ has weight $w_p \neq 1/2$, and the right mode has weight $1 - w_p$. Similarly, $w$ and $1 - w$ are the mode weights of $q$. Let $\mathbb{Q}_L$ and $\mathbb{Q}_R$ be the probability measures that respectively admit the density of the left and right modes of $q$, and $\mathbb{P}$ and $\mathbb{Q}_w$ be also the probability laws for $p$ and $q$.

**Assumption 2.2.** For distant bimodal mixture distributions $q$ and $p$ satisfying Assumption 2.1, and for $\eta \in (0, 1)$, we have $\left| \text{KSD}^2(\mathbb{P}, \mathbb{Q}_L)/\text{KSD}^2(\mathbb{P}, \mathbb{Q}_R) - 1 \right| < \eta$.

**Theorem 2.3.** Let $k_p$ be the Stein kernel associated with the radial kernel $k(\mathbf{x}, \mathbf{x}') = \phi(\|\mathbf{x} - \mathbf{x}'\|_2/\ell)$, where $\mathbf{x}, \mathbf{x}' \in \mathbb{R}^d$, $\ell > 0$, and $\phi \in \mathcal{C}^2(\mathbb{R})$, such that $\phi(z) \to 0$, $\phi'(z) \to 0$, and $\phi''(z) \to 0$ for $z \to \infty$. Let $p$ and $q$ be two bimodal mixture distributions satisfying Assumptions 2.1 and 2.2, for any $\eta \in (0, 1)$. We define $w^\star$ as the optimal mixture weight of $q$ with respect to the KSD distance, i.e., $w^\star = \underset{w \in [0,1]}{\text{argmin}} \, \text{KSD}(\mathbb{P}, \mathbb{Q}_w)$. Then, for $\mu$ large enough, we have $\left| w^\star - \frac{1}{2} \right| < \frac{\eta}{2(1-\eta)}$.

Table 3: AUCs obtained by Stein Thinning (ST) and Regularized Stein Thinning (RST). A 10-fold cross-validation is performed and the experiments are repeated 10 times to provide uncertainties.

| | NUTS Sampler | | | | | |
| --- | --- | --- | --- | --- | --- | --- |
| | $m = 50$ | | $m = 100$ | | $m = 300$ | |
| Dataset | ST | RST | ST | RST | ST | RST |
| Breast W. | 0.88 (0.020) | **0.96** (0.004) | 0.91 (0.023) | **0.96** (0.003) | 0.93 (0.008) | **0.96** (0.004) |
| Diabetes | **0.52** (0.009) | 0.50 (0.019) | 0.52 (0.021) | **0.55** (0.018) | 0.53 (0.015) | **0.57** (0.019) |
| Haberman | 0.51 (0.038) | **0.53** (0.023) | 0.54 (0.033) | **0.58** (0.035) | 0.53 (0.034) | **0.58** (0.017) |
| Liver | 0.53 (0.044) | **0.69** (0.014) | 0.56 (0.038) | **0.70** (0.013) | 0.61 (0.039) | **0.70** (0.011) |
| Sonar | 0.80 (0.021) | **0.81** (0.007) | 0.81 (0.009) | **0.82** (0.011) | 0.81 (0.011) | 0.81 (0.009) |

| | MALA Sampler | | | | | |
| --- | --- | --- | --- | --- | --- | --- |
| | $m = 50$ | | $m = 100$ | | $m = 300$ | |
| Dataset | ST | RST | ST | RST | ST | RST |
| Breast W. | 0.68 (0.044) | **0.93** (0.010) | 0.72 (0.048) | **0.93** (0.007) | 0.72 (0.037) | **0.88** (0.026) |
| Diabetes | **0.51** (0.012) | 0.48 (0.010) | **0.53** (0.028) | 0.51 (0.014) | 0.53 (0.016) | **0.56** (0.016) |
| Haberman | 0.52 (0.034) | **0.60** (0.027) | 0.53 (0.024) | **0.58** (0.017) | 0.55 (0.024) | **0.61** (0.013) |
| Liver | 0.54 (0.033) | **0.70** (0.008) | 0.55 (0.034) | **0.69** (0.005) | 0.57 (0.024) | **0.62** (0.032) |
| Sonar | 0.80 (0.019) | 0.80 (0.019) | 0.80 (0.010) | 0.80 (0.010) | **0.81** (0.013) | 0.80 (0.010) |

**Lemma 1.** *If $k_p$ is the Stein kernel associated with the radial kernel $k(\mathbf{x}, \mathbf{y}) = \phi(\|\mathbf{x} - \mathbf{y}\|_2/\ell)$, where $\ell > 0$, $\phi \in \mathcal{C}^2(\mathbb{R})$, and $\mathbf{x}, \mathbf{y} \in \mathbb{R}^d$ such that $s_p(\mathbf{x}), s_p(\mathbf{y}) < s_0$, then we have*

$$|k_p(\mathbf{x}, \mathbf{y})| \leq \frac{d-1}{\ell \|\mathbf{x} - \mathbf{y}\|_2} \phi'(\|\mathbf{x} - \mathbf{y}\|_2/\ell) + \frac{1}{\ell^2} \phi''(\|\mathbf{x} - \mathbf{y}\|_2/\ell) + \frac{2s_0}{\ell} \phi'(\|\mathbf{x} - \mathbf{y}\|_2/\ell)$$
$$+ s_0^2 \phi(\|\mathbf{x} - \mathbf{y}\|_2/\ell).$$

*Proof of Theorem B.* We consider the mixture distributions $p$ and $q$ satisfying Assumption 2.1, for $\mu > 0$ and $r > 0$, and assume that Assumption 2.2 is satisfied for $\eta \in (0, 1)$. More precisely, we denote by $q_L$ the distribution of the left mode of the mixture $q$, and similarly, $q_R$ is the distribution of the right mode of $q$. The probability measures $\mathbb{Q}_L$ and $\mathbb{Q}_R$ respectively admits the densities $q_L$ and $q_R$.

By definition of the KSD, we can write

$$\text{KSD}^2(\mathbb{P}, \mathbb{Q}_w) = \int k_p(\mathbf{x}, \mathbf{x}') q(\mathbf{x}) q(\mathbf{x}') d\mathbf{x} d\mathbf{x}'.$$

Additionally, given the above notations, $q$ takes the form $q = w q_L + (1 - w) q_R$. Then, we can develop the KSD expression to get

$$\text{KSD}^2(\mathbb{P}, \mathbb{Q}_w) = \int k_p(\mathbf{x}, \mathbf{x}')(w q_L(\mathbf{x}) + (1 - w) q_R(\mathbf{x}))(w q_L(\mathbf{x}') + (1 - w) q_R(\mathbf{x}')) d\mathbf{x} d\mathbf{x}'$$

$$= w^2 \int k_p(\mathbf{x}, \mathbf{x}') q_L(\mathbf{x}) q_L(\mathbf{x}') d\mathbf{x} d\mathbf{x}' + (1 - w)^2 \int k_p(\mathbf{x}, \mathbf{x}') q_R(\mathbf{x}) q_R(\mathbf{x}') d\mathbf{x} d\mathbf{x}'$$

$$+ 2w(1 - w) \int k_p(\mathbf{x}, \mathbf{x}') q_L(\mathbf{x}) q_R(\mathbf{x}') d\mathbf{x} d\mathbf{x}',$$

where the last term follows from the symmetry of $k_p$. Finally, we have

$$\text{KSD}^2(\mathbb{P}, \mathbb{Q}_w) = w^2 \text{KSD}^2(\mathbb{P}, \mathbb{Q}_L) + (1 - w)^2 \text{KSD}^2(\mathbb{P}, \mathbb{Q}_R)$$

$$+ 2w(1 - w) \int k_p(\mathbf{x}, \mathbf{x}') q_L(\mathbf{x}) q_R(\mathbf{x}') d\mathbf{x} d\mathbf{x}',$$

and we denote by $\Delta_{L,R}$ the last term of this equation, which now writes

$$\text{KSD}^2(\mathbb{P}, \mathbb{Q}_w) = w^2 \text{KSD}^2(\mathbb{P}, \mathbb{Q}_L) + (1-w)^2 \text{KSD}^2(\mathbb{P}, \mathbb{Q}_R) + 2w(1-w)\Delta_{L,R}. \qquad (4)$$

We first focus on the last term $\Delta_{L,R}$ of this expression, which can be shown to be arbitrarily small when $\mu$ gets large. According to Assumption 2.1, the distance between the centers of the two modes is $2\mu$, and both $q_L$ and $q_R$ have a compact support included in a ball of radius $r$. Consequently, for $\mathbf{x}, \mathbf{x}' \in \mathbb{R}^d$ such that $q_L(\mathbf{x}) > 0$ and $q_R(\mathbf{x}') > 0$, then $\|\mathbf{x} - \mathbf{x}'\|_2 > 2(\mu - r)$. Additionally, since the score $s_p$ is continuous, $s_p$ is bounded on a compact set, and it exists $s_0 > 0$ such that $s_p(\mathbf{x}) < s_0$ and $s_p(\mathbf{x}') < s_0$. Then, from Lemma 1, we have

$$|k_p(\mathbf{x}, \mathbf{x}')| \leq \frac{d-1}{2\ell(\mu - r)} \phi'(\|\mathbf{x} - \mathbf{x}'\|_2/\ell) + \frac{1}{\ell^2} \phi''(\|\mathbf{x} - \mathbf{x}'\|_2/\ell) + \frac{2s_0}{\ell} \phi'(\|\mathbf{x} - \mathbf{x}'\|_2/\ell)$$
$$+ s_0^2 \phi(\|\mathbf{x} - \mathbf{x}'\|_2/\ell),$$

and since $q_L(\mathbf{x})q_R(\mathbf{x}') = 0$ for $\|\mathbf{x} - \mathbf{x}'\|_2 < 2(\mu - r)$, we get

$$\Delta_{L,R} \leq \sup_{z > 2(\mu-r)/\ell} \left\{ \frac{d-1}{2\ell(\mu - r)} \phi'(z) + \frac{1}{\ell^2} \phi''(z) + \frac{2s_0}{\ell} \phi'(z) + s_0^2 \phi(z) \right\}.$$

By assumption, $\phi(z) \to 0$, $\phi'(z) \to 0$, and $\phi''(z) \to 0$ for $z \to \infty$, and then, we have

$$\lim_{\mu \to \infty} \Delta_{L,R} = 0.$$

Next, we reorder the terms of equation (4) to get a second-order polynomial in $w$ as follows

$$\text{KSD}^2(\mathbb{P}, \mathbb{Q}_w) = w^2 \big[ \text{KSD}^2(\mathbb{P}, \mathbb{Q}_L) + \text{KSD}^2(\mathbb{P}, \mathbb{Q}_R) - 2\Delta_{L,R} \big]$$
$$- 2w \big[ \text{KSD}^2(\mathbb{P}, \mathbb{Q}_R) - \Delta_{L,R} \big] + \text{KSD}^2(\mathbb{P}, \mathbb{Q}_R).$$

Notice that the coefficient of $w^2$ is $\text{KSD}^2(\mathbb{P}, \mathbb{Q}_{1/2})/4$, and is therefore positive. Then, $\text{KSD}^2(\mathbb{P}, \mathbb{Q}_w)$ admits a unique minimum with respect to $w$, given by

$$w^\star = \frac{\text{KSD}^2(\mathbb{P}, \mathbb{Q}_R) - \Delta_{L,R}}{\text{KSD}^2(\mathbb{P}, \mathbb{Q}_L) + \text{KSD}^2(\mathbb{P}, \mathbb{Q}_R) - 2\Delta_{L,R}}.$$

We rewrite $w^\star$ as follows,

$$w^\star = \frac{1/2\text{KSD}^2(\mathbb{P}, \mathbb{Q}_R) + 1/2\text{KSD}^2(\mathbb{P}, \mathbb{Q}_L) - \Delta_{L,R} + 1/2\text{KSD}^2(\mathbb{P}, \mathbb{Q}_R) - 1/2\text{KSD}^2(\mathbb{P}, \mathbb{Q}_L)}{\text{KSD}^2(\mathbb{P}, \mathbb{Q}_L) + \text{KSD}^2(\mathbb{P}, \mathbb{Q}_R) - 2\Delta_{L,R}}$$

$$= \frac{1}{2} + \frac{1}{2} \frac{\text{KSD}^2(\mathbb{P}, \mathbb{Q}_R) - \text{KSD}^2(\mathbb{P}, \mathbb{Q}_L)}{\text{KSD}^2(\mathbb{P}, \mathbb{Q}_L) + \text{KSD}^2(\mathbb{P}, \mathbb{Q}_R) - 2\Delta_{L,R}}$$

$$= \frac{1}{2} + \frac{1}{2} \frac{1 - \text{KSD}^2(\mathbb{P}, \mathbb{Q}_L)/\text{KSD}^2(\mathbb{P}, \mathbb{Q}_R)}{1 + \text{KSD}^2(\mathbb{P}, \mathbb{Q}_L)/\text{KSD}^2(\mathbb{P}, \mathbb{Q}_R) - 2\Delta_{L,R}/\text{KSD}^2(\mathbb{P}, \mathbb{Q}_R)}$$

$$= \frac{1}{2} + \frac{1}{2} \frac{1 - \text{KSD}^2(\mathbb{P}, \mathbb{Q}_L)/\text{KSD}^2(\mathbb{P}, \mathbb{Q}_R)}{2(1 - \Delta_{L,R}/\text{KSD}^2(\mathbb{P}, \mathbb{Q}_R)) + (\text{KSD}^2(\mathbb{P}, \mathbb{Q}_L)/\text{KSD}^2(\mathbb{P}, \mathbb{Q}_R) - 1)}.$$

We can deduce the following bound

$$\left| w^\star - \frac{1}{2} \right| \leq \frac{1}{2} \frac{|\text{KSD}^2(\mathbb{P}, \mathbb{Q}_L)/\text{KSD}^2(\mathbb{P}, \mathbb{Q}_R) - 1|}{|2(1 - \Delta_{L,R}/\text{KSD}^2(\mathbb{P}, \mathbb{Q}_R)) + (\text{KSD}^2(\mathbb{P}, \mathbb{Q}_L)/\text{KSD}^2(\mathbb{P}, \mathbb{Q}_R) - 1)|}. \qquad (5)$$

According to Assumption 2.2, with $0 < \eta < 1$,

$$\left| \frac{\text{KSD}^2(\mathbb{P}, \mathbb{Q}_L)}{\text{KSD}^2(\mathbb{P}, \mathbb{Q}_R)} - 1 \right| < \eta,$$

which gives an upper bound for the numerator of the right hand side of inequality (5). Additionally, for $\mu$ large enough, $\Delta_{R,L}$ is arbitrarily small, and in particular, we can have $\Delta_{R,L} < \text{KSD}^2(\mathbb{P}, \mathbb{Q}_R)/2$, and then $2(1 - \Delta_{L,R}/\text{KSD}^2(\mathbb{P}, \mathbb{Q}_R)) > 1$. Next, we use the triangle inequality to get

$$|2(1 - \Delta_{L,R}/\text{KSD}^2(\mathbb{P}, \mathbb{Q}_R)) + (\text{KSD}^2(\mathbb{P}, \mathbb{Q}_L)/\text{KSD}^2(\mathbb{P}, \mathbb{Q}_R) - 1)|$$
$$\geq 2(1 - \Delta_{L,R}/\text{KSD}^2(\mathbb{P}, \mathbb{Q}_R)) - |\text{KSD}^2(\mathbb{P}, \mathbb{Q}_L)/\text{KSD}^2(\mathbb{P}, \mathbb{Q}_R) - 1|$$
$$\geq 1 - \eta,$$

where the last inequality is obtained using $2(1 - \Delta_{L,R}/\text{KSD}^2(\mathbb{P}, \mathbb{Q}_R)) > 1$ for $\mu$ large enough, and Assumption 2.2 again. Finally, this lower bound on the denominator and the upper bound on the numerator combined with inequality (5) give

$$\left| w^\star - \frac{1}{2} \right| < \frac{\eta}{2(1-\eta)}.$$

$\square$

*Proof of Lemma 1.* We consider $\mathbf{x}, \mathbf{y} \in \mathbb{R}^d$, such that $\mathbf{x} \neq \mathbf{y}$, $s_p(\mathbf{x}) < s_0$ and $s_p(\mathbf{y}) < s_0$. From Equation (2) of the main article, we derive the Stein kernel obtained for a radial kernel $\phi(\|\mathbf{x} - \mathbf{y}\|_2/\ell)$, where $\ell > 0$ and $\phi \in \mathcal{C}^2(\mathbb{R})$, and get

$$k_p(\mathbf{x}, \mathbf{y}) = \frac{1-d}{\ell\|\mathbf{x} - \mathbf{y}\|_2}\phi'(\|\mathbf{x} - \mathbf{y}\|_2/\ell) - \frac{1}{\ell^2}\phi''(\|\mathbf{x} - \mathbf{y}\|_2/\ell)$$

$$- (s_p(\mathbf{x}) - s_p(\mathbf{y})) \cdot (\mathbf{x} - \mathbf{y})\frac{\phi'(\|\mathbf{x} - \mathbf{y}\|_2/\ell)}{\ell\|\mathbf{x} - \mathbf{y}\|_2} + (s_p(\mathbf{x}) \cdot s_p(\mathbf{y}))\phi(\|\mathbf{x} - \mathbf{y}\|_2/\ell).$$

Using Cauchy-Schwartz inequality, we have $s_p(\mathbf{x}) \cdot s_p(\mathbf{y}) \leq s_0^2$, and

$$\frac{(s_p(\mathbf{x}) - s_p(\mathbf{y})) \cdot (\mathbf{x} - \mathbf{y})}{\ell\|\mathbf{x} - \mathbf{y}\|_2} \leq \frac{2s_0}{\ell}.$$

Overall, we obtain the following bound

$$|k_p(\mathbf{x}, \mathbf{y})| \leq \frac{1-d}{\ell\|\mathbf{x} - \mathbf{y}\|_2}\phi'(\|\mathbf{x} - \mathbf{y}\|_2/\ell) + \frac{1}{\ell^2}\phi''(\|\mathbf{x} - \mathbf{y}\|_2/\ell) + \frac{2s_0}{\ell}\phi'(\|\mathbf{x} - \mathbf{y}\|_2/\ell)$$

$$+ s_0^2\phi(\|\mathbf{x} - \mathbf{y}\|_2/\ell).$$

$\square$

## C    Proofs of Theorem 2.4, Corollary 2.5, and Corollary 2.6

**Theorem 2.4** (KSD spurious minimum). *Let $k_p$ be the Stein kernel associated with the IMQ kernel with $\ell > 0$, $\beta \in (0,1)$, and $c = 1$. Let $\{\mathbf{x}_i\}_{i=1}^m \subset \mathcal{M}_{s_0} = \{\mathbf{x} \in \mathbb{R}^d : \|s_p(\mathbf{x})\|_2 \leq s_0\}$ be a fixed set of points of empirical measure $\mathbb{Q}_m = \frac{1}{m}\sum_{i=1}^m \delta(\mathbf{x}_i)$, with $s_0 \geq 0$ and $m \geq 2$. We have $\text{KSD}^2(\mathbb{P}, \mathbb{Q}_m) < \mathbb{E}[\text{KSD}^2(\mathbb{P}, \mathbb{P}_m)]$, if the score threshold $s_0$ and the sample size $m$ are small enough to satisfy $m < 1 + (\mathbb{E}[\|s_p(\mathbf{X})\|_2^2] - s_0^2)/(2\beta d/\ell^2 + 2\beta s_0/\ell + s_0^2)$.*

**Corollary 2.5** (Low KSD samples at density minimum). *Let $k_p$ be the Stein kernel associated with the IMQ kernel with $\ell > 0$, $\beta \in (0,1)$, and $c = 1$. Let $p$ be a density with at least one local minimum or saddle point. For $m \geq 2$, if $\{\mathbf{x}_i\}_{i=1}^m \subset \mathbb{R}^d$ is a set of points, all located at local minimum or saddle points of $p$, then we have $\text{KSD}^2(\mathbb{P}, \mathbb{Q}_m) < \mathbb{E}[\text{KSD}^2(\mathbb{P}, \mathbb{P}_m)]$, if $m < 1 + \frac{\ell^2}{2\beta d}\mathbb{E}[\|s_p(\mathbf{X})\|_2^2]$.*

**Corollary 2.6** (KSD spurious minimum for Gaussian mixtures). *Let $k_p$ be the Stein kernel associated with the IMQ kernel with $\ell > 0$, $\beta \in (0,1)$, and $c = 1$. Let the density $p$ be a Gaussian mixture model of two components with equal weights, respectively centered in $(-\mu, \mathbf{0}_{d-1})$ and $(\mu, \mathbf{0}_{d-1})$, of variance $\sigma^2\mathbf{I_d}$, and let $\nu = \mu/\sigma$. If $\nu > 1$ and $0 \leq s_0 < \left[\nu\sqrt{\nu^2 - 1} - \ln(\nu + \sqrt{\nu^2 - 1})\right]/\mu$, then for any $\{\mathbf{x}_i\}_{i=1}^m \subset \mathcal{M}_{s_0}$ of empirical measure $\mathbb{Q}_m$, we have*

*(i) $\text{KSD}^2(\mathbb{P}, \mathbb{Q}_m) < \mathbb{E}[\text{KSD}^2(\mathbb{P}, \mathbb{P}_m)]$ if $m$ and $s_0$ satisfy $m < 1 + \frac{\mathbb{E}[\|s_p(\mathbf{X})\|_2^2] - s_0^2}{2\beta d/\ell^2 + 2\beta s_0/\ell + s_0^2}$,*

*(ii) there exists three disjoint intervals $I_{-\mu}, I_0, I_\mu \subset \mathbb{R}$, respectively centered around $-\mu$, $0$, and $\mu$, such that $x_1^{(1)}, \ldots, x_m^{(1)} \in I_{-\mu} \cup I_0 \cup I_\mu$.*

**Lemma 2.** *Let $k_p$ be the Stein kernel associated to the IMQ kernel, with parameters $\ell > 0$, $\beta \in (0,1)$, and $c = 1$. For $s_0 \geq \min_{\mathbf{x} \in \mathbb{R}^d} \|s(\mathbf{x})\|_2$, and $\mathbf{x}, \mathbf{y} \in \mathcal{M}_{s_0}$, we have*

$$k_p(\mathbf{x}, \mathbf{y}) \leq \frac{2\beta d}{\ell^2} + \frac{2\beta s_0}{\ell} + s_0^2.$$

*Proof of Theorem 2.4.* By definition of the kernelized Stein discrepancy between the target distribution $\mathbb{P}$ and the empirical measure $\mathbb{P}_m = \frac{1}{m}\sum_{i=1}^m \delta(\mathbf{X}_i)$, we have

$$\mathbb{E}[\text{KSD}^2(\mathbb{P}, \mathbb{P}_m)] = \frac{1}{m^2}\sum_{i,j=1}^m \mathbb{E}[k_p(\mathbf{X}_i, \mathbf{X}_j)]$$

In what follows, $k_p$ denotes the Stein kernel obtained for the inverse multi-quadratric kernel function, *i.e.*,

$$k_p(\mathbf{x}, \mathbf{y}) = -\frac{4\beta(\beta+1)}{\ell^4}\|\mathbf{x} - \mathbf{y}\|_2^2(1 + \|\mathbf{x} - \mathbf{y}\|_2^2/\ell^2)^{-\beta-2}$$
$$+ \frac{2\beta}{\ell^2}(d + (s_p(\mathbf{x}) - s_p(\mathbf{y})) \cdot (\mathbf{x} - \mathbf{y}))(1 + \|\mathbf{x} - \mathbf{y}\|_2^2/\ell^2)^{-\beta-1}$$
$$+ (s_p(\mathbf{x}) \cdot s_p(\mathbf{y}))(1 + \|\mathbf{x} - \mathbf{y}\|_2^2/\ell^2)^{-\beta}.$$

Given that $\mathbf{X}_i$ and $\mathbf{X}_j$ are independent random variables that follow the distribution $\mathbb{P}$, one has $\mathbb{E}[k_p(\mathbf{X}_i, \mathbf{X}_j)] = 0$ for any $i \neq j$. Using this property and the closed-form expression of the Stein kernel $k_p$, it is found that

$$\mathbb{E}[\mathrm{KSD}^2(\mathbb{P}, \mathbb{P}_m)] = \frac{1}{m^2}\sum_{i=1}^{m}\mathbb{E}[k_p(\mathbf{X}_i, \mathbf{X}_i)]$$
$$= \frac{1}{m}\mathbb{E}[k_p(\mathbf{X}, \mathbf{X})]$$
$$= \frac{2\beta d}{m\ell^2} + \frac{\mathbb{E}[\|s_p(\mathbf{X})\|_2^2]}{m},$$

where $\mathbf{X} \sim \mathbb{P}$. On the other hand, the kernelized Stein discrepancy between the target $\mathbb{P}$ and the empirical distribution $\mathbb{Q}_m = \frac{1}{m}\sum_{i=1}^{m}\delta(\mathbf{x}_i)$ is given by

$$\mathrm{KSD}^2(\mathbb{P}, \mathbb{Q}_m) = \frac{1}{m^2}\sum_{i,j=1}^{m}k_p(\mathbf{x}_i, \mathbf{x}_j)$$
$$= \frac{1}{m^2}\sum_{i=1}^{m}k_p(\mathbf{x}_i, \mathbf{x}_i) + \frac{1}{m^2}\sum_{i \neq j}^{m}k_p(\mathbf{x}_i, \mathbf{x}_j)$$
$$= \frac{2\beta d}{m\ell^2} + \frac{1}{m^2}\sum_{i=1}^{m}\|s_p(\mathbf{x}_i)\|_2^2 + \frac{1}{m^2}\sum_{i \neq j}^{m}k_p(\mathbf{x}_i, \mathbf{x}_j).$$

Next, it can be shown that the difference $m\big(\mathrm{KSD}^2(\mathbb{P}, \mathbb{Q}_m) - \mathbb{E}[\mathrm{KSD}^2(\mathbb{P}, \mathbb{P}_m)]\big)$ takes the form

$$m\big(\mathrm{KSD}^2(\mathbb{P}, \mathbb{Q}_m) - \mathbb{E}[\mathrm{KSD}^2(\mathbb{P}, \mathbb{P}_m)]\big) = \frac{1}{m}\sum_{i=1}^{m}\|s_p(\mathbf{x}_i)\|_2^2 + \frac{1}{m}\sum_{i \neq j}^{m}k_p(\mathbf{x}_i, \mathbf{x}_j) - \mathbb{E}[\|s_p(\mathbf{X})\|_2^2].$$

Since $\mathbf{x}_i, \mathbf{x}_j \in \mathcal{M}_{s_0}$, we can use Lemma 2 to bound the terms $k_p(\mathbf{x}_i, \mathbf{x}_j)$, and then obtain

$$m\big(\mathrm{KSD}^2(\mathbb{P}, \mathbb{Q}_m) - \mathbb{E}[\mathrm{KSD}^2(\mathbb{P}, \mathbb{P}_m)]\big) \leq s_0^2 + (m-1)\Big(\frac{2\beta d}{\ell^2} + \frac{2\beta s_0 c_1}{\ell} + s_0^2\Big) - \mathbb{E}[\|s_p(\mathbf{X})\|_2^2],$$

where the right hand side is always negative if

$$m < 1 + \frac{\mathbb{E}[\|s_p(\mathbf{X})\|_2^2] - s_0^2}{2\beta d/\ell^2 + 2\beta s_0/\ell + s_0^2}.$$

$\square$

*Proof of Lemma 2.* The Stein kernel $k_p$ obtained for the inverse multi-quadratric kernel function, with parameters $\ell > 0$, $\beta \in (0, 1)$, and $c = 1$, is given by

$$k_p(\mathbf{x}, \mathbf{y}) = -\frac{4\beta(\beta+1)}{\ell^4}\|\mathbf{x} - \mathbf{y}\|_2^2(1 + \|\mathbf{x} - \mathbf{y}\|_2^2/\ell^2)^{-\beta-2}$$
$$+ \frac{2\beta}{\ell^2}(d + (s_p(\mathbf{x}) - s_p(\mathbf{y})) \cdot (\mathbf{x} - \mathbf{y}))(1 + \|\mathbf{x} - \mathbf{y}\|_2^2/\ell^2)^{-\beta-1}$$
$$+ (s_p(\mathbf{x}) \cdot s_p(\mathbf{y}))(1 + \|\mathbf{x} - \mathbf{y}\|_2^2/\ell^2)^{-\beta}.$$

Since the first term is always negative and $(1 + \|\mathbf{x} - \mathbf{y}\|_2^2/\ell^2)^{-\alpha} \leq 1$ for $\alpha = \beta, \beta + 1$, we obtain

$$k_p(\mathbf{x}, \mathbf{y}) \leq \frac{2\beta d}{\ell^2} + \frac{2\beta}{\ell^2}|(s_p(\mathbf{x}) - s_p(\mathbf{y})) \cdot (\mathbf{x} - \mathbf{y})|(1 + \|\mathbf{x} - \mathbf{y}\|_2^2/\ell^2)^{-\beta - 1} + |(s_p(\mathbf{x}) \cdot s_p(\mathbf{y}))|$$

$$\leq \frac{2\beta d}{\ell^2} + \frac{2\beta}{\ell}|(s_p(\mathbf{x}) - s_p(\mathbf{y})) \cdot \frac{\mathbf{x} - \mathbf{y}}{\|\mathbf{x} - \mathbf{y}\|_2}|\frac{\|\mathbf{x} - \mathbf{y}\|_2/\ell}{(1 + \|\mathbf{x} - \mathbf{y}\|_2^2/\ell^2)^{\beta + 1}} + |(s_p(\mathbf{x}) \cdot s_p(\mathbf{y}))|.$$

We define the function $g$ for $z \geq 0$ as

$$g(z) = \frac{z}{(1 + z^2)^{\beta + 1}},$$

and a simple function analysis shows that

$$\frac{c_1}{2} \overset{\text{def}}{=} \sup_{z \geq 0} g_1(z) = \frac{1}{\sqrt{2\beta + 1}} \left(\frac{2\beta + 1}{2\beta + 2}\right)^{\beta + 1}.$$

Since $\beta \in (0, 1)$, we have $c_1 \leq 1$. We combine the last two inequalities for $k_p(\mathbf{x}, \mathbf{y})$ and $c_1$ to get

$$k_p(\mathbf{x}, \mathbf{y}) \leq \frac{2\beta d}{\ell^2} + \frac{\beta}{\ell}|(s_p(\mathbf{x}) - s_p(\mathbf{y})) \cdot \frac{\mathbf{x} - \mathbf{y}}{\|\mathbf{x} - \mathbf{y}\|_2}| + |(s_p(\mathbf{x}) \cdot s_p(\mathbf{y}))|.$$

We can apply Cauchy-Schwartz inequality, and since $\mathbf{x}, \mathbf{y} \in \mathcal{M}_{s_0}$, we get

$$\left|(s_p(\mathbf{x}) - s_p(\mathbf{y})) \cdot \frac{\mathbf{x} - \mathbf{y}}{\|\mathbf{x} - \mathbf{y}\|_2}\right| \leq \|(s_p(\mathbf{x}) - s_p(\mathbf{y}))\|_2 \frac{\|\mathbf{x} - \mathbf{y}\|_2}{\|\mathbf{x} - \mathbf{y}\|_2} \leq 2s_0,$$

and also

$$|(s_p(\mathbf{x}) \cdot s_p(\mathbf{y}))| \leq s_0^2.$$

Overall, we obtain

$$k_p(\mathbf{x}, \mathbf{y}) \leq \frac{2\beta d}{\ell^2} + \frac{2\beta s_0}{\ell} + s_0^2.$$

$\square$

*Proof of Corollary 2.5.* As minimum and saddle points are stationary points of $p$, we have

$$\{\mathbf{x}_i\}_{i=1}^m \subset \mathcal{M}_0,$$

and we can apply Theorem 2.4 for $s_0 = 0$ to get the final result. $\square$

*Proof of Corollary 2.6.* Let the density $p$ be a Gaussian mixture model of two components with equal weights, respectively centered in $(-\mu, \mathbf{0}_{d-1})$ and $(\mu, \mathbf{0}_{d-1})$, and of variance $\sigma^2 \mathbf{I_d}$, and let $\nu = \mu/\sigma$. We assume that $\nu > 1$ and $0 \leq s_0 < [\nu\sqrt{\nu^2 - 1} - \ln(\nu + \sqrt{\nu^2 - 1})]/\mu$. Then, according to Theorem 2.4, for any $\{\mathbf{x}_i\}_{i=1}^m \subset \mathcal{M}_{s_0}$ of empirical measure $Q_m = \frac{1}{m}\sum_{i=1}^m \delta(\mathbf{x}_i)$, we have

(i) $\text{KSD}^2(\mathbb{P}, \mathbb{Q}_m) < \mathbb{E}[\text{KSD}^2(\mathbb{P}, \mathbb{P}_m)]$ if $m$ and $s_0$ satisfy $m < 1 + \frac{\mathbb{E}[\|s_p(\mathbf{X})\|_2^2] - s_0^2}{2\beta d/\ell^2 + 2\beta s_0/\ell + s_0^2}$.

To prove statement (ii), we need to characterize the shape of the set $\mathcal{M}_{s_0} \subset \mathbb{R}^d$, given by the level lines of the squared score norm $\|s_p(\mathbf{x})\|_2^2$. The density $p$ is a Gaussian mixture, i.e.,

$$p(\mathbf{x}) = \frac{1}{2(2\pi)^{d/2}\sigma^d}e^{-\|\mathbf{x}^{(-1)}\|_2^2/2\sigma^2}\left(e^{-(x^{(1)} + \mu)^2/2\sigma^2} + e^{-(x^{(1)} - \mu)^2/2\sigma^2}\right),$$

where $\mathbf{x}^{(-1)}$ is the vector $\mathbf{x}$ without the first component. Then, the score is also given by an explicit formula,

$$s_p(\mathbf{x}) = \begin{pmatrix} -\frac{x^{(1)}}{\sigma^2} + \frac{\mu}{\sigma^2}\tanh(\frac{\mu}{\sigma^2}x^{(1)}) \\ -\frac{\mathbf{x}^{(-1)}}{\sigma^2} \end{pmatrix},$$

where $\tanh$ is the standard hyperbolic tangent function. An important property of this score function is that the $j$-th component of $s_p$ only depends on $x^{(j)}$, which makes $s_p^{(j)}(\mathbf{x})$ invariant by any translation orthogonal to the $j$-th axis. Then, we can compute the squared score norm

$$\|s_p(\mathbf{x})\|_2^2 = s_p^{(1)}(x^{(1)})^2 + \frac{\|\mathbf{x}^{(-1)}\|_2^2}{\sigma^4},$$

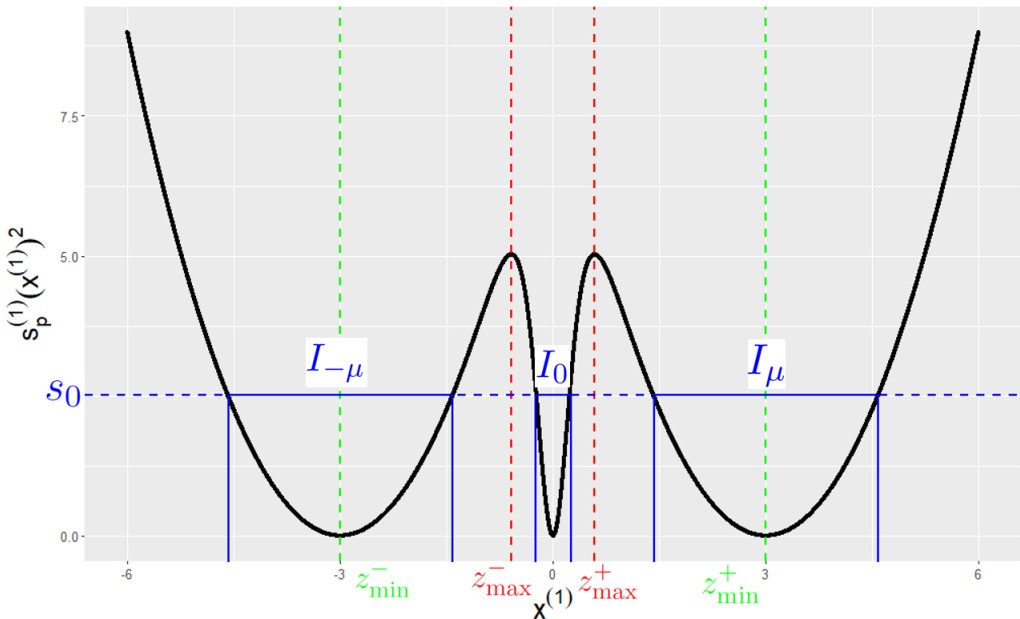

Figure 16: Squared first component of the score for a Gaussian mixture with $\mu = 3$ and $\sigma = 1$.

where $s_p^{(1)}(z)^2 = \left(\frac{z}{\sigma^2} - \frac{\mu}{\sigma^2}\tanh(\frac{\mu}{\sigma^2}z)\right)^2$. A simple function analysis of this univariate function, illustrated in Figure 16, shows that $s_p^{(1)}(z)^2$ has two local maximum in $z_{\max}^-$ and $z_{\max}^+$, and three local minimum in $z_{\min}^-$, $0$, and $z_{\min}^+$, provided that $\nu = \mu/\sigma > 1$. We also get that $s_p^{(1)}(z)^2$ grows to $+\infty$ when $x^{(1)} \to +/-\infty$. The extreme values are ordered as follows

$$-\mu < z_{\min}^- < z_{\max}^- < 0 < z_{\max}^+ < z_{\min}^+ < \mu.$$

The values of $z_{\min}^-$, $z_{\max}^-$, $z_{\max}^+$, and $z_{\min}^+$ are given by the zeros of the first derivative of $s_p^{(1)}(z)^2$, defined by

$$\frac{ds_p^{(1)}(z)^2}{dz} = 2\left(-\frac{1}{\sigma^2} + \left(\frac{\mu}{\sigma^2}\right)^2 \frac{1}{\cosh(\frac{\mu}{\sigma^2}z)^2}\right)\left(-\frac{z}{\sigma^2} + \frac{\mu}{\sigma^2}\tanh\left(\frac{\mu}{\sigma^2}z\right)\right).$$

This derivative vanishes when one of the two factors is null. Since $\mu/\sigma > 1$, the first term is null when

$$\frac{\mu^2}{\sigma^2}\frac{1}{\cosh(\frac{\mu}{\sigma^2}z)^2} = 1,$$

which leads to

$$z_{\max}^- = -\frac{\sigma^2}{\mu}\text{arcosh}\left(\frac{\mu}{\sigma}\right) \quad \text{and} \quad z_{\max}^+ = \frac{\sigma^2}{\mu}\text{arcosh}\left(\frac{\mu}{\sigma}\right).$$

The second factor is null when

$$\tanh\left(\frac{\mu}{\sigma^2}z\right) - \frac{z}{\mu} = 0. \tag{6}$$

Obviously, $z = 0$ is solution. Since $\mu/\sigma > 1$, equation (6) has two additional solutions. Although they do not have a closed form, we have

$$z_{\min}^- \in (-\mu, z_{\max}^-)$$
$$z_{\min}^+ \in (z_{\max}^+, \mu).$$

Also notice that, as $\mu/\sigma$ gets larger, $z_{\min}^-$ is closer to $-\mu$, and $z_{\min}^+$ to $\mu$. For example in Figure 16, we set $\mu/\sigma = 3$, and we hardly see a gap between $-\mu$ and $z_{\min}^-$, or $\mu$ and $z_{\min}^+$.

By definition, for $\mathbf{x} \in \mathcal{M}_{s_0}$, we have

$$s_p^{(1)}(x^{(1)})^2 \leq \|s_p(\mathbf{x})\|_2^2 \leq s_0^2.$$

Therefore, given the variations of $s_p^{(1)}(x^{(1)})^2$ detailed above and illustrated in Figure 16, if $s_0^2 < s_p^{(1)}(z_{\max}^-)^2 = s_p^{(1)}(z_{\max}^+)^2$, it exists three disjoint intervals $I_{-\mu}, I_0, I_\mu \subset \mathbb{R}$, respectively centered around $-\mu$, 0, and $\mu$, such that

$$\mathbf{x} \in \mathcal{M}_{s_0} \implies x^{(1)} \in I_{-\mu} \cup I_0 \cup I_\mu.$$

To conclude, we compute the value of $s_p^{(1)}(z_{\max}^-)$, that is

$$s_p^{(1)}(z_{\max}^+) = -\frac{1}{\mu}\text{arcosh}\left(\frac{\mu}{\sigma}\right) + \frac{\mu}{\sigma^2}\tanh\left(\text{arcosh}\left(\frac{\mu}{\sigma}\right)\right).$$

Using the formulas $\tanh(\text{arcosh}(x)) = \frac{\sqrt{x^2-1}}{x}$ for $|x| > 1$, $\text{arcosh}(x) = \ln(x + \sqrt{x^2-1})$, and with $\nu = \mu/\sigma$, we get

$$s_p^{(1)}(z_{\max}^+) = \sqrt{(\mu/\sigma)^2 - 1}/\sigma - \ln(\mu/\sigma + \sqrt{(\mu/\sigma)^2 - 1})/\mu$$
$$= \left[\nu\sqrt{\nu^2 - 1} - \ln(\nu + \sqrt{\nu^2 - 1})\right]/\mu,$$

which is always strictly positive since $\nu > 1$. By assumption, $0 \leq s_0 < \left[\nu\sqrt{\nu^2 - 1} - \ln(\nu + \sqrt{\nu^2 - 1})\right]/\mu$, and therefore, we have $s_0 < s_p^{(1)}(z_{\max}^+) = s_p^{(1)}(z_{\max}^-)$, which concludes the proof of statement (ii).

$\qquad\square$

## D   Proof of Theorem 3.3

**Theorem 3.3.** *Let $k_p$ be the Stein kernel associated with the radial kernel $k(\mathbf{x}, \mathbf{x}') = \phi(\|\mathbf{x} - \mathbf{x}'\|_2/\ell)$, where $\mathbf{x}, \mathbf{x}' \in \mathbb{R}^d$, $\ell > 0$, and $\phi \in \mathcal{C}^2(\mathbb{R})$. Let $p$ and $q$ be two bimodal mixture distributions satisfying Assumption 2.1. We define $w_\lambda^\star$ as the optimal mixture weight of $q$ with respect to the entropic regularized KSD distance, i.e., $w_\lambda^\star = \underset{w \in [0,1]}{\text{argmin}}\ \text{KSD}_\lambda(\mathbb{P}, \mathbb{Q}_w)$. If $\mathbb{E}[\log(p(\mathbf{Z}_L))] \neq \mathbb{E}[\log(p(\mathbf{Z}_R))]$ where $\mathbf{Z}_L \sim \mathbb{Q}_L$ and $\mathbf{Z}_R \sim \mathbb{Q}_R$, it exists $\lambda \in \mathbb{R}$ such that $w_\lambda^\star = w_p$.*

*Proof of Theorem 3.3.* We consider the mixture distributions $p$ and $q$ satisfying Assumption 2.1, for $\mu > 0$ and $r > 0$, and denote by $q_L$ the distribution of the left mode of the mixture $q$, and similarly, $q_R$ is the distribution of the right mode of $q$. The probability measures $\mathbb{Q}_L$ and $\mathbb{Q}_R$ respectively admits the densities $q_L$ and $q_R$. As in the proof of Theorem 2.3, we get that

$$\text{KSD}^2(\mathbb{P}, \mathbb{Q}_w) = w^2\left[\text{KSD}^2(\mathbb{P}, \mathbb{Q}_L) + \text{KSD}^2(\mathbb{P}, \mathbb{Q}_R) - 2\Delta_{L,R}\right]$$
$$- 2w\left[\text{KSD}^2(\mathbb{P}, \mathbb{Q}_R) - \Delta_{L,R}\right] + \text{KSD}^2(\mathbb{P}, \mathbb{Q}_R).$$

Next, we define $\mathbf{Z} \sim \mathbb{Q}_w$, $\mathbf{Z}_L \sim \mathbb{Q}_L$, and $\mathbf{Z}_R \sim \mathbb{Q}_R$, and develop the entropic regularization term,

$$\mathbb{E}[\log(p(\mathbf{Z}))] = \int \log(p(\mathbf{x}))(wq_L(\mathbf{x}) + (1-w)q_R(\mathbf{x}))d\mathbf{x}$$
$$= w\int \log(p(\mathbf{x}))q_L(\mathbf{x})d\mathbf{x} + (1-w)\int \log(p(\mathbf{x}))q_R(\mathbf{x})d\mathbf{x}$$
$$= w\mathbb{E}[\log(p(\mathbf{Z}_L))] + (1-w)\mathbb{E}[\log(p(\mathbf{Z}_R))].$$

Combining these two results, we have

$$\text{KSD}_\lambda^2(\mathbb{P}, \mathbb{Q}_w) = w^2\left[\text{KSD}^2(\mathbb{P}, \mathbb{Q}_L) + \text{KSD}^2(\mathbb{P}, \mathbb{Q}_R) - 2\Delta_{L,R}\right]$$
$$- 2w\left[\text{KSD}^2(\mathbb{P}, \mathbb{Q}_R) - \Delta_{L,R} + \lambda/2(\mathbb{E}[\log(p(\mathbf{Z}_L))] - \mathbb{E}[\log(p(\mathbf{Z}_R))])\right]$$
$$+ \text{KSD}^2(\mathbb{P}, \mathbb{Q}_R) - \lambda\mathbb{E}[\log(p(\mathbf{Z}_R))].$$

We recall that $p$ does not depend on $w$, but only on the fixed weight $w_p$ and the two mode distributions. Consequently, $\text{KSD}_\lambda^2(\mathbb{P}, \mathbb{Q}_w)$ is a second-order polynomial with respect to $w$. As for Theorem 2.3, the

coefficient of $w^2$ is $\text{KSD}^2(\mathbb{P}, \mathbb{Q}_{1/2})/4$, and is therefore positive. Then, the polynomial is minimized with respect to $w$, at $w^\star$ defined by

$$w_\lambda^\star = \frac{\text{KSD}^2(\mathbb{P}, \mathbb{Q}_R) - \Delta_{L,R} + \lambda/2(\mathbb{E}[\log(p(\mathbf{Z}_L))] - \mathbb{E}[\log(p(\mathbf{Z}_R))])}{\text{KSD}^2(\mathbb{P}, \mathbb{Q}_L) + \text{KSD}^2(\mathbb{P}, \mathbb{Q}_R) - 2\Delta_{L,R}}.$$

Since $\mathbb{E}[\log(p(\mathbf{Z}_L))] - \mathbb{E}[\log(p(\mathbf{Z}_R))] \neq 0$ by assumption, we get that $w_\lambda^\star = w_p$ for lambda defined by

$$\lambda = 2\frac{w_p\text{KSD}^2(\mathbb{P}, \mathbb{Q}_L) - (1 - w_p)\text{KSD}^2(\mathbb{P}, \mathbb{Q}_R) + (1 - 2w_p)\Delta_{L,R}}{\mathbb{E}[\log(p(\mathbf{Z}_L))] - \mathbb{E}[\log(p(\mathbf{Z}_R))]}.$$

$\square$

# E  Proof of Theorem 3.4

**Theorem 3.4.** *Let $k_p$ be the Stein kernel associated with the IMQ kernel with $\ell > 0$, $\beta \in (0, 1)$, and $c = 1$. For $m \geq 2$, let $\{\mathbf{x}_i\}_{i=1}^m \subset \mathbb{R}^d$ be a set of points concentrated at $\mathbf{x}_0$, a local minimum or saddle point of $p$, and of empirical measure $\mathbb{Q}_m$. Then, we have $\text{L-KSD}^2(\mathbb{P}, \mathbb{Q}_m) > \mathbb{E}[\text{L-KSD}^2(\mathbb{P}, \mathbb{P}_m)]$, if the density at $\mathbf{x}_0$ satisfies $p(\mathbf{x}_0) < \Delta^+ p(\mathbf{x}_0)/(\mathbb{E}[\|s_p(\mathbf{X})\|_2^2] + \mathbb{E}[\Delta^+ \log p(\mathbf{X})])$.*

*Proof of Theorem 3.4.* Following the same approach as in the proof of Theorem 2.4, we have

$$\mathbb{E}[\text{L-KSD}^2(\mathbb{P}, \mathbb{P}_m)] = \frac{2\beta d}{m\ell^2} + \frac{\mathbb{E}[\|s_p(\mathbf{X})\|_2^2]}{m} + \frac{\mathbb{E}[\Delta^+ \log p(\mathbf{X})]}{m},$$

where $\mathbf{X} \sim \mathbb{P}$, and, with the empirical distribution $\mathbb{Q}_m = \frac{1}{m}\sum_{i=1}^m \delta(\mathbf{x}_i)$,

$$\text{L-KSD}^2(\mathbb{P}, \mathbb{Q}_m) = \frac{2\beta d}{m\ell^2} + \frac{1}{m^2}\sum_{i=1}^m \|s_p(\mathbf{x}_i)\|_2^2 + \frac{1}{m^2}\sum_{i\neq j}^m k_p(\mathbf{x}_i, \mathbf{x}_j) + \frac{1}{m^2}\sum_{i=1}^m \Delta^+ \log(p(\mathbf{x}_i)).$$

As $\{\mathbf{x}_i\}_{i=1}^m$ are concentrated at a local minimum or saddle point $\mathbf{x}_0$, the score is null for all particles, as well as the distances between them, and we get

$$\text{L-KSD}^2(\mathbb{P}, \mathbb{Q}_m) = \frac{2\beta d}{m\ell^2} + \frac{(m-1)2\beta d}{m\ell^2} + \frac{\Delta^+ \log(p(\mathbf{x}_0))}{m}.$$

Next, the difference $m(\text{L-KSD}^2(\mathbb{P}, \mathbb{Q}_m) - \mathbb{E}[\text{L-KSD}^2(\mathbb{P}, \mathbb{P}_m)])$ writes

$$m(\text{L-KSD}^2(\mathbb{P}, \mathbb{Q}_m) - \mathbb{E}[\text{L-KSD}^2(\mathbb{P}, \mathbb{P}_m)]) = \Delta^+ \log(p(\mathbf{x}_0)) + (m-1)\frac{2\beta d}{\ell^2}$$
$$- (\mathbb{E}[\|s_p(\mathbf{X})\|_2^2] + \mathbb{E}[\Delta^+ \log p(\mathbf{X})]).$$

By definition,

$$\Delta^+ \log p(\mathbf{x}) \overset{def}{=} \sum_{j=1}^d \left(\frac{\partial^2 \log p(\mathbf{x})}{\partial x^{(j)2}}\right)^+, \tag{7}$$

with

$$\frac{\partial^2 \log p(\mathbf{x})}{\partial x^{(j)2}} = \frac{1}{p(\mathbf{x})}\frac{\partial^2 p(\mathbf{x})}{\partial x^{(j)2}} - \left(\frac{1}{p(\mathbf{x})}\frac{\partial p(\mathbf{x})}{\partial x^{(j)}}\right)^2.$$

As $\mathbf{x}_0$ is a stationary point of $p$, $\partial p(\mathbf{x}_0)/\partial x^{(j)} = 0$, and we obtain

$$\frac{\partial^2 \log p(\mathbf{x}_0)}{\partial x^{(j)2}} = \frac{1}{p(\mathbf{x}_0)}\frac{\partial^2 p(\mathbf{x}_0)}{\partial x^{(j)2}},$$

leading to

$$\Delta^+ \log p(\mathbf{x}_0) = \sum_{j=1}^d \frac{1}{p(\mathbf{x}_0)}\left(\frac{\partial^2 p(\mathbf{x}_0)}{\partial x^{(j)2}}\right)^+ = \frac{\Delta^+ p(\mathbf{x}_0)}{p(\mathbf{x}_0)}. \tag{8}$$

Finally, we get

$$m\big(\text{L-KSD}^2(\mathbb{P}, \mathbb{Q}_m) - \mathbb{E}[\text{L-KSD}^2(\mathbb{P}, \mathbb{P}_m)]\big) = \frac{\Delta^+ p(\mathbf{x}_0)}{p(\mathbf{x}_0)} + (m-1)\frac{2\beta d}{\ell^2}$$
$$- \big(\mathbb{E}[\|s_p(\mathbf{X})\|_2^2] + \mathbb{E}[\Delta^+ \log p(\mathbf{X})]\big),$$

which ensures that $\text{L-KSD}^2(\mathbb{P}, \mathbb{Q}_m) - \mathbb{E}[\text{L-KSD}^2(\mathbb{P}, \mathbb{P}_m)] > 0$ for $m \geq 2$, provided that

$$p(\mathbf{x}_0) < \frac{\Delta^+ p(\mathbf{x}_0)}{\mathbb{E}[\|s_p(\mathbf{X})\|_2^2] + \mathbb{E}[\Delta^+ \log p(\mathbf{X})]}.$$

$\square$

# F   Proof of Theorem 3.6

**Assumption 3.5.** Let $\mathbb{Q}$ be a probability distribution on $\mathbb{R}^d$, such that $\mathbb{P}$ is absolutely continuous with respect to $\mathbb{Q}$. Let $\{\mathbf{Z}_i\}_{i \in \mathbb{N}} \subset \mathbb{R}^d$ be a $\mathbb{Q}$-invariant, time-homogeneous Markov chain, generated using a $V$-uniformly ergodic transition kernel, such that $V(\mathbf{x}) \geq \frac{d\mathbb{P}}{d\mathbb{Q}}\sqrt{2\beta d/\ell^2 + \|s_p(\mathbf{x})\|_2^2}$. Suppose that, for some $\gamma > 0$, $\sup_{i \in \mathbb{N}} \mathbb{E}[e^{\gamma|\log(p(\mathbf{Z}_i))|}] < \infty$, $\sup_{i \in \mathbb{N}} \mathbb{E}[e^{\gamma \Delta^+ \log p(\mathbf{Z}_i)}] < \infty$,

$$\sup_{i \in \mathbb{N}} \mathbb{E}\big[e^{\gamma \max(1, \frac{d\mathbb{P}}{d\mathbb{Q}}(\mathbf{Z}_i)^2)(\frac{2\beta d}{\ell^2} + \|s_p(\mathbf{Z}_i)\|_2^2)}\big] < \infty, \quad \sup_{i \in \mathbb{N}} \mathbb{E}\Big[\frac{d\mathbb{P}}{d\mathbb{Q}}(\mathbf{Z}_i)\sqrt{\frac{2\beta d}{\ell^2} + \|s_p(\mathbf{Z}_i)\|_2^2} V(\mathbf{Z}_i)\Big] < \infty.$$

Assumption 3.5 is close to the assumption made in Riabiz et al. [2022]. For the last two finite expectations of the assumption, notice that we have just plugged the formula $k_p(\mathbf{x}, \mathbf{x}) = 2\beta d/\ell^2 + \|s_p(\mathbf{x})\|^2$ into the integrability conditions, since this formula is quite straightforward. But in addition to Riabiz et al. [2022], we require two integrability assumptions, one for each regularization term. We give additional insights about these quite complex conditions below. Overall, our assumptions are hardly stronger than those of Riabiz et al. [2022].

The condition $E[e^{\gamma|\log(p(\mathbf{Z}_i))|}] < \infty$ is satisfied if $\int f(\mathbf{x})p(\mathbf{x})^{-\gamma}dx < \infty$, where $f$ is the distribution of $\mathbf{Z}_i$ (since $p(\mathbf{x}) > 1$ only on a compact set and $p$ is continuous). Therefore, there exists $\gamma > 0$ satisfying such condition, provided that the tails of density $f$ are not too heavy compared to the tails of $p$. This has to be verified for all iterations of the Markov Chain, and happens to be a quite mild assumption.

Regarding the condition involving the Laplacian term, we can use the analysis from Riabiz et al. (2022) (Appendix S2.4). It relies on a Lipschitz condition for the score function $\nabla \log p$, to transform the first finite expectation of Assumption 3.5 into a more amenable and practical integrability condition which writes $E[\kappa\|\mathbf{Z}_i\|_2^2] < \infty$ for some $\kappa \in (0, \infty)$. In the same spirit, if we add a Lipschitz condition on $\Delta^+ \log p$, our second additional integrability assumption reduces to a similar amenable condition. Finally, even though Riabiz et al (2022) do not elaborate further on this condition, it can be noticed that it is satisfied if all $\mathbf{Z}_i$ have subgaussian tails, for example. Besides, one of the main example of distantly dissipative distributions are log-concave functions outside of a compact set. In this case, the Laplacian regularization is null outside of a compact set, since the Laplacian is negative for concave functions. Then, the integrability condition is automatically satisfied for this type of distributions.

**Theorem 3.6.** *Let $\mathbb{P}$ be a distantly dissipative probability measure, that admits the density $p \in \mathcal{C}^2(\mathbb{R}^d)$, $k_p$ be the Stein kernel associated with the IMQ kernel wh $\ell, c > 0, \beta \in (0, 1)$. Let $\{\mathbf{Z}_i\}_{i \in \mathbb{N}} \subset \mathbb{R}^d$ be a Markov chain satisfying Assumption 3.5, $\pi$ be the index sequence of length $m_n$ generated by regularized Stein thinning, and $\mathbb{Q}_{m_n}$ be the empirical measure of $\{\mathbf{Z}_{\pi_i}\}_{i=1}^{m_n}$. If $\log(n)^\alpha < m_n < n$, with any $\alpha > 1$, and $\lambda_{m_n} = o(\log(m_n)/m_n)$, then we have almost surely $\mathbb{Q}_{m_n} \underset{n \to \infty}{\Longrightarrow} \mathbb{P}$.*

Theorem 3.6 extends Theorem 3 from Riabiz et al. [2022] to regularized Stein Thinning, using Lemmas 3-4 and assuming the following convergence rate of the regularization parameter $\lambda_{m_n} = o(\log(m_n)/m_n)$. Lemma 3 also extends Theorem 1 from Riabiz et al. [2022][Theorem 1], whereas Lemma 4 is a slight modification of Lemma 5 from Riabiz et al. [2022].

**Lemma 3.** *Let $\mathbb{P}$ be a probability measure on $\mathbb{R}^d$ that admits density $p \in \mathcal{C}^2(\mathbb{R}^d)$, $k_p$ be a reproducing Stein kernel, and $\{\mathbf{x}_i\}_{i=1}^n \subset \mathbb{R}^d$ a fixed set of points. If $\pi$ is an index sequence of length $m$ produced by regularized Stein thinning, then we have for $\lambda > 0$,*

$$\mathrm{KSD}^2\big(\frac{1}{m}\sum_{j=1}^m \delta(\mathbf{x}_{\pi_j})\big) \leq \mathrm{KSD}^2\big(\sum_{i=1}^n w_i^\star \delta(\mathbf{x}_i)\big) + \frac{1+\log(m)}{m} \max_{i=1,\ldots,n} k_p(\mathbf{x}_i, \mathbf{x}_i)$$

$$+ \frac{1+\log(m)}{m} \max_{i=1,\ldots,n} \Delta^+ \log(p(\mathbf{x}_i)) + 2\lambda \max_{i=1,\ldots,n} |\log(p(\mathbf{x}_i))|,$$

*where the weights $w^\star$ are defined as*

$$w^\star \in \arg\min_{\substack{\sum_i w_i = 1 \\ w_i \geq 0}} \mathrm{KSD}^2\big(\sum_{i=1}^n w_i \delta(\mathbf{x}_i)\big).$$

**Lemma 4.** *Let $f$ be a non-negative function on $\mathbb{R}^d$. Consider a sequence of random variables $(\mathbf{X}_i)_{i \in \mathbb{N}} \subset \mathbb{R}^d$ such that, for some $\gamma > 0$,*

$$b \overset{def}{=} \sup_{i \in \mathbb{N}} \mathbb{E}[e^{\gamma f(\mathbf{X}_i)}] < \infty.$$

*If $\log(n)^\alpha < m_n < n$, with any $\alpha > 1$, then we have almost surely,*

$$\lim_{n \to \infty} \frac{\log(m_n)}{m_n} \max_{i=1,\ldots,n} f(\mathbf{X}_i) = 0.$$

The proofs of Lemmas 3 and 4 are reported at the end of this section. We first proceed with the proof of Theorem 3.6.

*Proof of Theorem 3.6.* From Lemma 3, we have

$$\mathrm{KSD}^2\big(\frac{1}{m_n}\sum_{j=1}^{m_n} \delta(\mathbf{Z}_{\pi_j})\big) \leq \underbrace{\mathrm{KSD}^2\big(\sum_{i=1}^n w_i^\star \delta(\mathbf{Z}_i)\big)}_{(\star)} + \underbrace{\frac{1+\log(m_n)}{m_n} \max_{i=1,\ldots,n} k_p(\mathbf{Z}_i, \mathbf{Z}_i)}_{(\star\star)}$$

$$+ \underbrace{\frac{1+\log(m_n)}{m_n} \max_{i=1,\ldots,n} \Delta^+ \log(p(\mathbf{Z}_i))}_{(\star\star\star)} + \underbrace{2\lambda_{m_n} \max_{i=1,\ldots,n} |\log(p(\mathbf{Z}_i))|}_{(\diamond)}.$$

Riabiz et al. [2022][Proof of Theorem 3, p.12] showed that the term $(\star)$ converges towards 0 almost surely as $n \to \infty$. For the remaining terms, from Assumption 3.5, we have

$$\sup_{i \in \mathbb{N}} \mathbb{E}[e^{\gamma(d/\ell^2 + \|s_p(\mathbf{Z}_i)\|_2^2)}] < \infty, \quad \sup_{i \in \mathbb{N}} \mathbb{E}[e^{\gamma|\log(p(\mathbf{Z}_i))|}] < \infty,$$

and

$$\sup_{i \in \mathbb{N}} \mathbb{E}[e^{\gamma \Delta^+ \log(p(\mathbf{Z}_i))}] < \infty.$$

We can use Lemma 4 with $f(\mathbf{x}) = k_p(\mathbf{x}, \mathbf{x})$ and $f(\mathbf{x}) = \Delta^+ \log(\mathbf{x})$ to deduce that $(\star\star) \to 0$ and $(\star\star\star) \to 0$, respectively. The remaining term $(\diamond)$ can be rewritten as

$$2\lambda_{m_n} \max_{i=1,\ldots,n} |\log(p(\mathbf{Z}_i))| = \frac{2m_n\lambda_{m_n}}{\log(m_n)} \times \frac{\log(m_n)}{m_n} \max_{i=1,\ldots,n} |\log(p(\mathbf{Z}_i))|.$$

Using the assumption that $\lambda_{m_n} = o(\log(m_n)/m_n)$ and Lemma 4 with $f(\mathbf{x}) = |\log(p(\mathbf{x}))|$, we conclude that $(\diamond) \to 0$. It follows that $\mathrm{KSD}^2\big(\frac{1}{m_n}\sum_{j=1}^{m_n} \delta(\mathbf{x}_{\pi_j})\big) \to 0$ almost surely as $n \to \infty$. Given that $p$ is assumed to be distantly dissipative, we apply Theorem 4 from Chen et al. [2019] to obtain that $\mathbb{Q}_{m_n} \Rightarrow \mathbb{P}$ almost surely, as $n \to \infty$. $\qquad\square$

*Proof of Lemma 3.* We consider an iteration $t \in \{2, \ldots, m\}$ of regularized Stein thinning, where $m \geq 2$ is the final length of the thinned sample. We define $a_t = t^2 \text{KSD}^2(\mathbb{P}, \frac{1}{t} \sum_{j=1}^{t} \delta(\mathbf{x}_{\pi_j}))$ and $f_t = \sum_{j=1}^{t} k_p(\mathbf{x}_{\pi_j}, \cdot)$. We also denote $S_1^2 = \max_{i=1,\ldots,n} k_p(\mathbf{x}_i, \mathbf{x}_i) + \max_{i=1,\ldots,n} \Delta^+ \log(p(\mathbf{x}_i))$, and $S_2 = 2 \max_{i=1,\ldots,n} |\log(p(\mathbf{x}_i))|$. Using the definition of the squared KSD of an empirical measure, we have

$$a_t = a_{t-1} + k_p(\mathbf{x}_{\pi_t}, \mathbf{x}_{\pi_t}) + 2 \sum_{j=1}^{t-1} k_p(\mathbf{x}_{\pi_j}, \mathbf{x}_{\pi_t}).$$

Let $\mathbf{x}_t^\star = \arg\min_{\mathbf{y} \in \{\mathbf{x}_i\}_{i=1}^n} f_{t-1}(\mathbf{y})$. By definition, $\mathbf{x}_{\pi_t}$ minimizes the cost function of the regularized Stein thinning algorithm at iteration $t$, and we have

$$k_p(\mathbf{x}_{\pi_t}, \mathbf{x}_{\pi_t}) + 2 \sum_{j=1}^{t-1} k_p(\mathbf{x}_{\pi_j}, \mathbf{x}_{\pi_t}) + \Delta^+ \log(p(\mathbf{x}_{\pi_t})) - \lambda t \log(p(\mathbf{x}_{\pi_t}))$$
$$\leq k_p(\mathbf{x}_t^\star, \mathbf{x}_t^\star) + 2 \sum_{j=1}^{t-1} k_p(\mathbf{x}_{\pi_j}, \mathbf{x}_t^\star) + \Delta^+ \log(p(\mathbf{x}_t^\star)) - \lambda t \log(p(\mathbf{x}_t^\star)).$$

We combine this last inequality with the first equation to obtain

$$a_t \leq a_{t-1} + k_p(\mathbf{x}_t^\star, \mathbf{x}_t^\star) + 2 \sum_{j=1}^{t-1} k_p(\mathbf{x}_{\pi_j}, \mathbf{x}_t^\star) + \Delta^+ \log(p(\mathbf{x}_t^\star)) - \Delta^+ \log(p(\mathbf{x}_{\pi_t}))$$
$$- \lambda t (\log(p(\mathbf{x}_t^\star)) - \log(p(\mathbf{x}_{\pi_t}))),$$

and then,

$$a_t \leq a_{t-1} + k_p(\mathbf{x}_t^\star, \mathbf{x}_t^\star) + 2 \sum_{j=1}^{t-1} k_p(\mathbf{x}_{\pi_j}, \mathbf{x}_t^\star) + \Delta^+ \log(p(\mathbf{x}_t^\star)) - \Delta^+ \log(p(\mathbf{x}_{\pi_t}))$$
$$+ \lambda t (|\log(p(\mathbf{x}_t^\star))| + |\log(p(\mathbf{x}_{\pi_t}))|).$$

By definition, $0 \leq |\log(p(\mathbf{x}_i))| \leq S_2$ and $k_p(\mathbf{x}_t^\star, \mathbf{x}_t^\star) + \Delta^+ \log(p(\mathbf{x}_t^\star)) - \Delta^+ \log(p(\mathbf{x}_{\pi_t})) \leq S_1^2$, hence, we have

$$a_t \leq a_{t-1} + S_1^2 + t\lambda S_2 + 2 \min_{\mathbf{y} \in \{\mathbf{x}_i\}_{i=1}^n} f_{t-1}(\mathbf{y}).$$

As in the proof of Riabiz et al. [2022], we have $\min_{\mathbf{y} \in \{\mathbf{x}_i\}_{i=1}^n} f_{t-1}(\mathbf{y}) \leq \sqrt{a_{t-1}} \|h^\star\|_{\mathcal{H}(k_p)}$ where $h^\star$ is the element in the RKHS $\mathcal{H}(k_p)$ of the form $h^\star = \sum_{i=1}^{n} w_i^\star k_p(\mathbf{x}_i, \cdot)$. As a result, $a_t$ is bounded as follows:

$$a_t \leq a_{t-1} + S_1^2 + t\lambda S_2 + 2\sqrt{a_{t-1}} \|h^\star\|_{\mathcal{H}(k_p)}.$$

We then show by induction that

$$a_t \leq t^2 (\|h^\star\|_{\mathcal{H}(k_p)}^2 + C_t + \lambda S_2),$$

where

$$C_t \stackrel{def}{=} \frac{1}{t} \left( S_1^2 - \|h^\star\|_{\mathcal{H}(k_p)}^2 \right) \sum_{j=1}^{t} \frac{1}{j}.$$

With such as a result, we will have $\text{KSD}^2(\mathbb{P}, \frac{1}{t} \sum_{j=1}^{t} \delta(\mathbf{x}_{\pi_j})) \leq \text{KSD}^2(\mathbb{P}, \sum_{i=1}^{n} w_i^\star \delta(\mathbf{x}_{\pi_j})) + C_t + \lambda S_2$ and obtain the advertised result in Theorem 3 for the last iteration $t = m$.

For $t = 1$, we have $a_1 = k_p(\mathbf{x}_{\pi_1}, \mathbf{x}_{\pi_1}) \leq S_1^2$ and thus $a_1 \leq \|h^\star\|_{\mathcal{H}(k_p)}^2 + C_1 + \lambda S_2$. For a fixed $t \geq 2$, assume that $a_{t-1} \leq (t-1)^2 (\|h^\star\|_{\mathcal{H}(k_p)}^2 + C_{t-1} + \lambda S_2)$ where $C_{t-1} = \frac{1}{t-1}(S_1^2 - \|h^\star\|_{\mathcal{H}(k_p)}^2) \sum_{j=1}^{t-1} \frac{1}{j}$. We then have

$$\begin{aligned}
a_t &\leq a_{t-1} + S_1^2 + t\lambda S_2 + 2\sqrt{a_{t-1}} \|h^\star\|_{\mathcal{H}(k_p)} \\
&\leq (t-1)^2 (\|h^\star\|_{\mathcal{H}(k_p)}^2 + C_{t-1} + \lambda S_2) + S_1^2 + t\lambda S_2 \\
&\qquad + 2(t-1)\sqrt{\|h^\star\|_{\mathcal{H}(k_p)}^2 + C_{t-1} + \lambda S_2} \|h^\star\|_{\mathcal{H}(k_p)} \\
&= t^2 (\|h^\star\|_{\mathcal{H}(k_p)}^2 + C_t + \lambda S_2) + R_t
\end{aligned} \tag{9}$$

where

$$R_t = (t-1)^2 C_{t-1} - t^2 C_t + (1-2t)(\|h^\star\|^2_{\mathcal{H}(k_p)} + \lambda S_2) + S_1^2$$
$$+ t\lambda S_2 + 2(t-1)\sqrt{\|h^\star\|^2_{\mathcal{H}(k_p)} + C_{t-1} + \lambda S_2}\|h^\star\|_{\mathcal{H}(k_p)}$$
$$= (t-1)^2 C_{t-1} - t^2 C_t + (1-2t)\|h^\star\|^2_{\mathcal{H}(k_p)} + S_1^2$$
$$+ \lambda S_2(1-t) + 2(t-1)\sqrt{\|h^\star\|^2_{\mathcal{H}(k_p)} + C_{t-1} + \lambda S_2}\|h^\star\|_{\mathcal{H}(k_p)}$$

Using Riabiz et al. [2022][Lemma 1], we have

$$2\|h^\star\|_{\mathcal{H}(k_p)}\sqrt{\|h^\star\|^2_{\mathcal{H}(k_p)} + C_{t-1} + \lambda S_2} \leq 2\|h^\star\|^2_{\mathcal{H}(k_p)} + C_{t-1} + \lambda S_2$$

It follows from Equation (9) that we need $R_t \leq 0$, *i.e.*,

$$2\|h^\star\|^2_{\mathcal{H}(k_p)} + C_{t-1} + \lambda S_2 \leq \frac{t^2 C_t - (t-1)^2 C_{t-1}}{t-1} - \frac{S_1^2 - \|h^\star\|^2_{\mathcal{H}(k_p)}}{t-1} + \lambda S_2 + 2\|h^\star\|^2_{\mathcal{H}(k_p)}.$$

The above inequality is always satisfied as long as

$$2\|h^\star\|^2_{\mathcal{H}(k_p)} + C_{t-1} + \leq \frac{t^2 C_t - (t-1)^2 C_{t-1}}{t-1} - \frac{S_1^2 - \|h^\star\|^2_{\mathcal{H}(k_p)}}{t-1} + 2\|h^\star\|^2_{\mathcal{H}(k_p)},$$

which is equivalent to

$$tC_t - (t-1)C_{t-1} \geq \frac{1}{t}(S_1^2 - \|h^\star\|^2_{\mathcal{H}(k_p)}),$$

and always true by definition of $C_t$. Hence we have shown that $a_t \leq t^2(\|h^\star\|^2_{\mathcal{H}(k_p)} + C_t + \lambda S_2)$. Given that $\|h^\star\|^2_{\mathcal{H}(k_p)} = \text{KSD}^2(\mathbb{P}, \sum_{i=1}^n w_i^\star \delta(\mathbf{x}_i))$, we have

$$\text{KSD}^2(\mathbb{P}, \frac{1}{t}\sum_{j=1}^t \delta(\mathbf{x}_{\pi_j})) \leq \text{KSD}^2(\mathbb{P}, \sum_{i=1}^n w_i^\star \delta(\mathbf{x}_i)) + C_t + \lambda S_2\,,$$

where $C_t \leq \frac{1+\log(t)}{t}\big(\max_{i=1,\dots,n} k_p(\mathbf{x}_i, \mathbf{x}_i) + \max_{i=1,\dots,n} \Delta^+ \log(p(\mathbf{x}_i))\big)$ (see [Riabiz et al., 2022][Lemma 2]). $\qquad\square$

*Proof of Lemma 4.* We follow the proof of Lemma 5 from [Riabiz et al., 2022]. In the last step of the proof, we essentially need to show that

$$\sum_{m_n=1}^\infty c_1(m_n^2) < \infty \quad\text{and}\quad \sum_{m_n=1}^\infty c_2(m_n) < \infty$$

where

$$c_1(m_n^2) \overset{def}{=} \frac{2\log(m_n)}{m_n^2}\frac{\log(nb)}{\gamma} \quad\text{and}\quad c_2(m_n) \overset{def}{=} 4\frac{\log(m_n)}{m_n^2}\frac{\log(n((m_n+1)^2)b)}{\gamma}\,.$$

With the assumption that $\log(n)^\beta \leq m_n < n$ with $\beta > 1$, it is deduced that $c_1(m_n^2) \to 0$ and $c_2(m_n) \to 0$ as $n \to \infty$. $\qquad\square$

## G   Laplacian Stein Operator

Instead of the standard Langevin operator, we can use $\mathcal{T}_p(g) = \Delta(pg)/p$, mentioned in Oates et al. [2017, Appendix A.2]. However, such Stein operator introduces similar problems as Pathology I, since the Stein kernel associated with $\mathcal{T}_p'(g)$ also has spurious minimum in regions where second derivatives of $p$ vanish, as illustrated below. Therefore, it is more appropriate to taylor a specific Laplacian correction as proposed in this paper, which cannot directly be derived from a Stein kernel, since it is not differentiable. In this appendix, we study the operator $\mathcal{T}_p$ defined as $\mathcal{T}_p g = \Delta(pg)/p$ and such that [Oates et al., 2017, Appendix A.2]

$$\mathbb{E}[(\mathcal{T}_p g)(\mathbf{Z})] = 0\,,$$

for all $g$ belonging to $\mathcal{G}$, and with $\mathbf{Z} \sim \mathbb{P}$. For each dimension $j \in \{1, \ldots, d\}$, let $\mathcal{T}_p^j$ be the operator defined as

$$(\mathcal{T}_p^j g)(\mathbf{x}) = \frac{1}{p(\mathbf{x})} \left( \nabla_{x_j}^2 p(\mathbf{x}) g(\mathbf{x}) + 2 \nabla_{x_j} p(\mathbf{x}) \nabla_{x_j} g(\mathbf{x}) + p(\mathbf{x}) \nabla_{x_i}^2 g(\mathbf{x}) \right)$$

for $g : \mathbb{R}^d \to \mathbb{R}$, and where $x_j$ is the $j$-th coordinate of $\mathbf{x}$ to simplify notations. The operator $\mathcal{T}_p g$ can then be rewritten as $(\mathcal{T}_p g)(\mathbf{x}) = \sum_{j=1}^d (\mathcal{T}_p^j g)(\mathbf{x})$. Gorham and Mackey [2017][Proposition 2] establishes the closed-form expression of the KSD in the case of the multidimensional Langevin operator. We generalize the proof of Gorham and Mackey [2017][Proposition 2] for the Laplacian operator $(\mathcal{T}_p g)(\mathbf{x}) = \Delta(p(\mathbf{x})g(\mathbf{x}))/p(\mathbf{x})$ and establish a closed-form expression of the Stein kernel $k_p : \mathbb{R}^d \times \mathbb{R}^d \to \mathbb{R}$. For a given kernel function $k : \mathbb{R}^d \times \mathbb{R}^d \to \mathbb{R}$, $k \in \mathcal{C}^{2,2}$, the Stein kernel $k_p$ is given by [Gorham and Mackey, 2017]

$$k_p(\mathbf{x}, \mathbf{y}) = \sum_{j=1}^d k_p^j(\mathbf{x}, \mathbf{y}), \tag{10}$$

where

$$k_p^j(\mathbf{x}, \mathbf{y}) = \langle \mathcal{T}_p^j(k(\mathbf{x}, \cdot)), \mathcal{T}_p^j(k(\cdot, \mathbf{y})) \rangle_{\mathcal{H}_k}. \tag{11}$$

After a few developments, it is found that

$$
\begin{aligned}
p(\mathbf{x})p(\mathbf{y})k_p^j(\mathbf{x}, \mathbf{y}) = {} & \nabla_{x_j}^2 p(\mathbf{x}) \nabla_{y_j}^2 p(\mathbf{y}) k(\mathbf{x}, \mathbf{y}) \\
& + 2\nabla_{x_j}^2 p(\mathbf{x}) \nabla_{y_j} p(\mathbf{y}) \nabla_{y_j} k(\mathbf{x}, \mathbf{y}) + p(\mathbf{y}) \nabla_{x_j}^2 p(\mathbf{x}) \nabla_{y_j}^2 k(\mathbf{x}, \mathbf{y}) \\
& + 2\nabla_{x_j} p(\mathbf{x}) \nabla_{y_j}^2 p(\mathbf{y}) \nabla_{x_j} k(\mathbf{x}, \mathbf{y}) + 4\nabla_{x_j} p(\mathbf{x}) \nabla_{y_j} p(\mathbf{y}) \nabla_{x_i} \nabla_{y_j} k(\mathbf{x}, \mathbf{y}) \\
& + 2p(\mathbf{y}) \nabla_{x_j} p(\mathbf{x}) \nabla_{x_j} \nabla_{y_j}^2 k(\mathbf{x}, \mathbf{y}) + p(\mathbf{x}) \nabla_{y_j}^2 p(\mathbf{y}) \nabla_{x_j}^2 k(\mathbf{x}, \mathbf{y}) \\
& + 2p(\mathbf{x}) \nabla_{y_j} p(\mathbf{y}) \nabla_{x_j}^2 \nabla_{y_j} k(\mathbf{x}, \mathbf{y}) + p(\mathbf{x})p(\mathbf{y}) \nabla_{x_j}^2 \nabla_{y_j}^2 k(\mathbf{x}, \mathbf{y})
\end{aligned}
\tag{12}
$$

A closed-form expression can then be obtained in the case of, *e.g.*, an inverse multiquadratic kernel of the form $k(\mathbf{x}, \mathbf{y}) = (1 + \|\mathbf{x} - \mathbf{y}\|_2^2/\ell^2)^{-1/2}$ where $\ell$ denotes the bandwidth. In this case, one has

$$
\begin{aligned}
& \nabla_{y_j} k(\mathbf{x}, \mathbf{y}) = \frac{1}{\ell^2} k(\mathbf{x}, \mathbf{y})^3 (x_j - y_j), \quad \nabla_{x_j} k(\mathbf{x}, \mathbf{y}) = -\nabla_{y_j} k(\mathbf{x}, \mathbf{y}) \\
& \nabla_{y_j}^2 k(\mathbf{x}, \mathbf{y}) = \nabla_{x_j}^2 k(\mathbf{x}, \mathbf{y}) = -\frac{1}{\ell^2} k(\mathbf{x}, \mathbf{y})^3 + \frac{3}{\ell^4} k(\mathbf{x}, \mathbf{y})^5 (x_j - y_j)^2 \\
& \nabla_{x_j} \nabla_{y_j} k(\mathbf{x}, \mathbf{y}) = \frac{1}{\ell^2} k(\mathbf{x}, \mathbf{y})^3 - \frac{3}{\ell^4} k(\mathbf{x}, \mathbf{y})^5 (x_j - y_j)^2 \\
& \nabla_{x_j}^2 \nabla_{y_j} k(\mathbf{x}, \mathbf{y}) = \frac{-9k(\mathbf{x}, \mathbf{y})^5}{\ell^4} (x_j - y_j) + \frac{15k(\mathbf{x}, \mathbf{y})^7}{\ell^6} (x_j - y_j)^3 \\
& \nabla_{x_j} \nabla_{y_j}^2 k(\mathbf{x}, \mathbf{y}) = -\nabla_{x_j}^2 \nabla_{y_j} k(\mathbf{x}, \mathbf{y}) \\
& \nabla_{x_j}^2 \nabla_{y_j}^2 k(\mathbf{x}, \mathbf{y}) = \frac{9k(\mathbf{x}, \mathbf{y})^5}{\ell^4} - \frac{90k(\mathbf{x}, \mathbf{y})^7}{\ell^6} (x_j - y_j)^2 + \frac{105k(\mathbf{x}, \mathbf{y})^9}{\ell^8} (x_j - y_j)^4
\end{aligned}
\tag{13}
$$

A closed-form expression of $k_p$ can then be obtained by combining Equations (10)-(13). In contrast to the Langevin Stein kernel (see Equation (2) of the main article), the above Stein kernel involves second-order derivatives of the density $p$.

We run experiments based on our Example 1 of Gaussian mixtures for this new Stein operator $\mathcal{T}_p g = \Delta(pg)/p$. We sequentially set $\mu = 2$ and $\mu = 5$, with $\sigma = 1$ and $w = 0.5$. Results are displayed in Figure 17. In the left panel with $\mu = 2$, we see that Pathology II does not occur, as opposed to Figure 2 of the main article with the Langevin Stein operator. However, when $\mu$ is set to 5 in the right panel, all particles are concentrated in spurious minimum again. Therefore, introducing higher-order derivatives of the target density through the Stein operator does not seem to be a promising route.

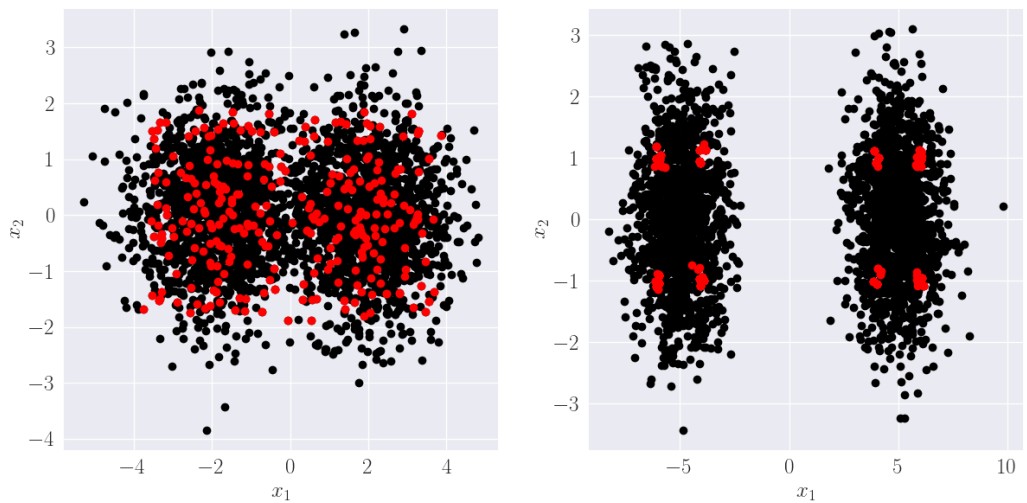

Figure 17: Stein thinning with the Stein operator $\mathcal{T}_p g = \Delta(pg)/p$, for Gaussian mixtures of Example 1, with $\mu = 2$ (left panel), and $\mu = 5$ (right panel), $\sigma = 1$, and $w = 0.5$.

