# OpenReview forum: "Kernel Stein Discrepancy thinning: a theoretical perspective of pathologies and a practical fix with regularization"
_NeurIPS.cc/2023/Conference — NeurIPS 2023 poster_

### Official Review · Reviewer_GM6g · 2023-06-21

**Soundness:** 4 excellent
**Presentation:** 3 good
**Contribution:** 3 good
**Rating:** 7
**Confidence:** 2

**Summary:**

This paper explores two pathologies associated with KSD thinning: the inability to distinguish mixing weights and concentration on low-probability regions of the target. The authors provide formalisations of when these pathologies can occur (Theorem 2.3 and Theorem 2.4), supported by empirical evidence. To address these issues, they propose the addition of two penalty terms: entropy regularisation and a Laplacian correction. The effectiveness of the proposed regularised KSD approach in mitigating these pathologies is demonstrated through both theoretical studies and numerical experiments.

**Strengths:**

**Novelty**: Although the pathologies of blindness to mixing proportions and spurious concentration to low-probability regions have been acknowledged in the community, this paper stands out by providing formalizations and solutions to these issues. Solutions to Pathology I has been previously proposed in other applications of KSD (e.g., Zhang et al., 2022; Liu et al., 2023), but no solution nor formalisation have been provided regarding Pathology II. The proposed regularisation for Stein thinning represents a novel and principled solution, supported by strong theoretical guarantees.

**Applicability**: The paper thoroughly discusses and provides experimental evidence on the impact of the regularisation parameter, the degree of freedom, on the performance of the proposed approach. Practical guidelines are given to aid in choosing appropriate parameter values in practice.

**Writing**: This paper is well motivated and clearly written.

Zhang, M., Key, O., Hayes, P., Barber, D., Paige, B., & Briol, F. X. (2022). Towards healing the blindness of score matching. *arXiv preprint arXiv:2209.07396*.

Liu, X., Duncan, A. B., & Gandy, A. (2023). Using Perturbation to Improve Goodness-of-Fit Tests based on Kernelized Stein Discrepancy. ICML 2023.

**Weaknesses:**

This paper is well written overall. I only found a couple of minor areas that could potentially be improved:

**Limitations**: It would be beneficial to include more in-depth discussions on any limitations of the L-KSD approach. E.g., whilst a convergence guarantee was given when $n \to \infty$ (Theorem 3.5), for finite $n$ and $m_n$, how prominent is the bias introduced due to the addition of regularisation?

**Clarity**: Whilst this paper is mostly clearly written, some parts deserve elaboration and clarification. E.g.,

1. In Section 3.2, the intuition behind why the entropic regularisation solves Pathology I is not very clearly explained (also see Q1).
2. In Theorem 3.3, it would be helpful to clarify the precise meaning of “concentrated at $x_0$”.
3. Also in Theorem 3.3, it might be beneficial to provide an interpretation of the inequality for $p(x_0)$ on L313. Specifically, is this condition satisfied by the mixture of Gaussian example used in the paper?
4. In Section 4, it would be helpful to clarify the choice of the kernel (and any hyper-parameters) in the MMD used to evaluate the results.

**Questions:**

**Q1. Entropic regularisation**: Could you elaborate on why the entropic regularisation helps alleviate Pathology I? In particular, L242 says “The main idea of this entropic regularization is that $− \log(p(x))$ takes higher values in modes of smaller probability, and therefore provides the relative mode weight information”. Whilst I agree with the first half of this claim, I am unsure how this sensitivity to low-density regions relates to the ability to detect mis-specification of the mixing weights, and why this regularisation would lead to small values when the mixing weights are correctly specified.

**Q2. Figure 3**: The proportion of particles selected by L-KSD that lie in the left mode, while better complying with the true proportion of $w=0.2$ compared to the result of KSD, still differs from the true value by a statistically significant gap (since the reported estimated proportion is $0.11 \pm 0.03$). Could you explain whether this gap is expected? Does it mean that the regularised penalty suffers from a significant bias in the finite sample scheme, despite being consistent in the infinite limit?

**Limitations:**

More discussions on any limitations of the approach should be discussed; see the question in "Weaknesses".

---

> ### Author Rebuttal · Authors · 2023-08-04
>
> Thank you again for the detailed review, and the questions raised. We anwser to the main points below. See also the global rebuttal for additional insights.
>
> **Theorem 3.3.** Regarding Theorem 3.3, “concentrated at $x_0$” means that all particles are located at $x_0$. We will improve clarity of this statement in the final version of the article. Thank you for the comment. The inequality of $p(x_0)$ tells us that the Laplacian regularization penalizes particles located around stationary points of the distribution in low probability regions. Since several intricate terms in this inequality depend on the distribution $p$, it is unfortunately not possible to give a precise interpretation of the threshold. However, many experiments show that such regularization is indeed efficient to avoid Stein thinning solutions with particles located in low probability regions. In particular, this is true for Example 1 with Gaussian mixtures. While it is possible to derive formula in this specific case (in the same spirit than Corollary 2.6), their interpretation is difficult.
>
> **Entropic regularization.** Modes with smaller weights take smaller density values, and are therefore more penalized by $–\log(p(x))$ than modes of higher weights. Therefore, with such entropic penalization, regularized Stein thinning tends to select particles in modes of higher weights more frequently than in modes of smaller weights, and we recover appropriate proportions.
>
> **Mode proportions in Figure 3 and finite sample bias.** The mode proportion achieved by L-KSD, although much better than the standard KSD one, is indeed not the true proportion in this example. This is not due to a significant bias from the regularized penalty, but this is related to our heuristic for $\lambda$. As mentioned in the global answer, we only have the guarantee that an optimal $\lambda$ exists to recover the true proportion, but unfortunately, a precise tuning is impossible, and our rule of thumb does not ensure such exact recovery. It only gives a default value, which consistently outperforms the standard KSD, with much closer proportions to the target distribution, more representative of the true balance between the modes.
>
> **MMD settings.** All details about the MMD settings are given in Appendix 1.2. We will add these details in the main paper thanks to the additional page if the paper is accepted.

---

> > ### Comment · Reviewer_GM6g · 2023-08-12
> >
> > I thank the authors for their responses, which have answered my questions. I would like to keep the scores.

---

### Official Review · Reviewer_XEkq · 2023-07-06

**Soundness:** 3 good
**Presentation:** 3 good
**Contribution:** 3 good
**Rating:** 7
**Confidence:** 3

**Summary:**

In this article the author(s) studied the Stein thining method, an algorithm for post-processing outputs of MCMC based on the kernelized Stein discrepancy (KSD). This article first theoretically analyzed two pathologies of KSD, and then proposed methods to mitigate the two issues by regularizing the KSD objective function. Both theoretical analysis and numerical experiments show that the regularized Stein thinning method has improved the sampling quality.

**Strengths:**

The blindness of score-based methods to multimodal distributions is a long-standing issue, and this paper presents a careful theoretical anaysis in the context of Stein thining. The author(s) also point out another pathology of KSD caused by spurious minimum of the objective function. Both analyses deepen our understanding of KSD-based methods. The proposed regularzed Stein thinning method also shows promising performance.

**Weaknesses:**

1. For the theoretical analyses, some questions need to be clarified. See the questions section.
2. For the numerical experiments, the target densities explored so far are somewhat simplified. I would suggest considering the following additional scenarios:
    - Sampling from energy-based models, e.g. [1].
    - Gaussian mixtures with more than two distant modes, e.g. the experiments in [2].
    - Bayesian neural networks, as an extension of the linear logistic regression.

These are all "difficult" target distributions, and I would not expect regularized Stein thinning solving them perfectly. But exploring at least  some of the cases above would be helpful.

3. If my understanding is correct, the truncated Laplacian operator requires computing the eigen decomposition of the Hessian matrix for every point in the MCMC sample, and this seems to be a huge computational cost. (After author rebuttal: **this point was my misunderstanding**, and the actual computing cost was clarified by the author(s). The truncated Laplacian operator only requires computing the diagonal elements of the Hessian matrix.)

[1] Che, T., Zhang, R., Sohl-Dickstein, J., Larochelle, H., Paull, L., Cao, Y., & Bengio, Y. (2020). Your GAN is secretly an energy-based model and you should use discriminator driven latent sampling. Advances in Neural Information Processing Systems.

[2] Qiu, Y., & Wang, X. (2023). Efficient Multimodal Sampling via Tempered Distribution Flow. Journal of the American Statistical Association.

**Questions:**

1. Both Theorem 2.3 and Theorem 3.2 are based on radial basis function kernels, whereas Theorem 2.4 and Theorem 3.3 are using  inverse multi-quadratic kernels. What considerations is this based on?
2. Assumption 2.2 seems too hard to verify. Can you give an example for it to hold? Considering Example 1 of this article with only one dimension, is it possible to given an expression for $\eta$? I would expect $\eta$ to be a function of $\mu$ and $w$, and $\eta\rightarrow 0$ as $\mu\rightarrow\infty$.
3. As mentioned in the weakness section, the computing time may be taken into account and reported in the experiments.

**Limitations:**

After rebuttal: my previous concern on computational cost is no longer a major problem, and other limitations may include the tuning of $\lambda$, and some technicalities such as the choice of kernel functions.

---

> ### Author Rebuttal · Authors · 2023-08-04
>
> Thank you again for the detailed review, and the many suggestions provided. We hope that we tackle the main points in the global answer above. We answer to the remaining specific points below.
>
> **Assumption 2.2.** Notice that Assumption 2.2 is satisfied when the two modes of $q$ have a similar KSD distance with respect to $p$. In particular, this is true for symmetric distributions. Unfortunately, it is not possible to give an expression for $\eta$, since it strongly depends on the distributions $p$ and $q$, and the result is stated for any continuous distributions $p$ and $q$.
>
> **Additional experiments.** Actually, we conducted unreported experiments with Gaussian mixtures with several distant modes as in [2] and in a setting similar to the first energy model considered in [1]. They show thinned samples of good quality even for such difficult target distributions, as you can see in the attached pdf in the global rebuttal. We will add such cases in the article to show the good behavior of the regularized Stein thinning. Thank you for this suggestion.
>
> Another suggestion is to conduct experiments with Bayesian neural networks, which is definitely an exciting idea. However, the scope of such application is very large, involves ultra-high dimensional distributions, and thus deserves a full separate paper in our opinion, since this setting has not been considered yet in the previous KSD literature.

---

> > ### Comment · Reviewer_XEkq · 2023-08-12
> >
> > I would like to thank the author(s) for their rebuttal that addresses most of my concerns. The clarification on computational cost (apologies for the previous misunderstanding) and the additional experiments better demonstrate the strength of the proposed method, and hence I have raised my score.
> >
> > I have one additional mild comment that the author(s) may consider to further improve the quality of the article: Can the author(s) also show the average proportion of the particles in each mode in the first figure of rebuttal PDF, similar to Figure 3? I feel the smaller modes tend to be over-shrunk. Does tuning $\lambda$ help to approach the correct mode weights?

---

> > > ### Author Response · Authors · 2023-08-12
> > >
> > > Thank you for reconsidering your score. You are right about the proportions in the first figure of the rebuttal PDF. We found that approximatively 6% of the particles are located in each of the smaller modes. We made a few experiments and found that is it possible to get closer to 10% by slightly reducing the value of lambda. With lambda = 0.4/m, a run gives weights of 0.103 and 0.083 in the smaller modes, for instance.
> > >
> > > Thank you again for your questions and suggestions.

---

### Official Review · Reviewer_zNe7 · 2023-07-21

**Soundness:** 3 good
**Presentation:** 4 excellent
**Contribution:** 3 good
**Rating:** 7
**Confidence:** 3

**Summary:**

This paper proposes a regularized version of Stein thinning for post-processing the output of Markov Chain Monte Carlo algorithms. The regularization addresses two common challenges of Stein thinning: insensitivity to mode proportions and samples concentrating at stationary points. The paper analyzes the above two pathologies under a theoretical framework which is then utilized to justify the proposed regularization technique. Numerical experiments demonstrate gains over standard Stein thinning in both toy examples and in a logistic regression setting.

**Strengths:**

- The paper is well-written and easy to follow. The structure of the paper is logical.

- I found the study of pathologies illuminating and interesting. The theoretical framework appears to be sound.

- The proposed technique shows promise both in the toy case and in the simple logistic regression setup.

**Weaknesses:**

- More intuition on the assumptions of the theoretical formulation would be helpful, with some discussion on the limitations of the results with more focus on practical applications of the method.

- Extra hyperparameter is introduced, for which a heuristic tuning method is provided, however a systematic approach is missing.

**Questions:**

- Why do we see an initial sharp increase of MMD with respect to $d$ followed by gradual decrease in the Gaussian mixture example (Fig. 5) for RST, but strict decrease in MMD for ST?

- Is there a systematic way to set $\lambda$ beyond the heuristics provided in the paper?

**Limitations:**

Limitations are not discussed.

---

> ### Author Rebuttal · Authors · 2023-08-04
>
> Thank you again for the review, and the questions raised. Notice that we tackle the main points in the global answer above.
>
> **MMD variations.** The variations of the MMD with respect to dimension d in Figure 5 indeed depend on the considered examples, with different observed behaviors for the Gaussian mixture and the banana-shaped one. This is quite difficult to interpret, since these variations are due to the intrinsic complexity of summarizing a high-dimensional distribution with just m=300 points (as in Fig. 5), and is thus case dependent. But we agree that it would be interesting to investigate more precisely what distribution properties could imply such differences.

---

> > ### Comment · Reviewer_zNe7 · 2023-08-16
> >
> > Thank you for the additional clarification. I raised my score to accept.

---

### Official Review · Reviewer_4eLX · 2023-07-26

**Soundness:** 3 good
**Presentation:** 3 good
**Contribution:** 3 good
**Rating:** 7
**Confidence:** 3

**Summary:**

The goal of Stein thinning is to post-process the outputs of Markov chain Monte Carlo (MCMC) methods by minimizing the kernelized Stein discrepancy (KSD) between the produced chain and the target distribution. It is useful and has become fast a quite popular method in Bayesian inference because it automatically removes the burn-in period, corrects the bias, and has asymptotic properties of convergence towards the target.

However it suffers from two pathologies, namely Pathology I of mode proportion blindness and Pathology II of mode collapse. Pathology I originates from the score insensitivity to mode weights, while Pathology II arises from the over-representation of a few modes. To mitigate these pathologies, the paper proposes a regularization technique called regularized Stein thinning based on adding an entropy and Laplacian term to the KSD.

**Strengths:**

The paper does a good pedagogic job on the issues related to the optimization of the KSD, and offers a complete picture by providing theoretical negative results about the regular KSD (eg Theorem 2.3, 2.4).
For instance Th 2.4 (through Cor 2.5) says one can attain a better KSD than empirical samples of the true target mixture by putting all the points between the modes because this is a region where the score is close to zero.

Moreover the proposed regularised KSD is shown to enjoy good properties regarding Pathologies I and II and has theoretical guarantees and extensive experiments to demonstrate its efficiency.

**Weaknesses:**

Theorem 2.4 is still limited to the IMQ kernel.

The Laplacian correction involves second-order derivatives of the target.

Their regularised KSD introduces an additional hyperparameter lambda that a user should tune, however it is not clear how to fix this parameter in advance (depending on the dimension, the target etc). It is not clear in the theoretical result Theorem 3.2 neither what lambda recovers the true mixture weights and fixes Pathology I. Still, the authors conducted a reasonable number of experiments with different values of lambda.

**Questions:**

Do the authors have any intuition on a general recipee or dependence of lambda on the parameters of the problem?

**Limitations:**

The authors reasonably discuss the limitations of their work.

---

> ### Author Rebuttal · Authors · 2023-08-04
>
> Thank you again for the review and the positive comments. We hope that we adress the identified weaknesses in the global answer above.

---

> > ### Comment · Reviewer_4eLX · 2023-08-11
> > **Response to rebutal**
> >
> > I read the authors' rebutal and other reviewers' comment. Other reviewers seem to share some of my concerns (e.g. extension of some theoretical results to other radial kernels, tuning of lambda) and eventually others (experimental settings not considered in the submission).
> > The authors rebutal reasonably adressed my concerns and their additional experiments on mixture of Gaussians advocate for their regularized Stein thinning. I still have a positive opinion on the paper and maintain my score.

---

### Author Rebuttal · Authors · 2023-08-04

We greatly thank the reviewers for their positive comments about our article and relevant suggestions. We explain below how we will improve the article clarity following the reviewer guidelines. We tackle the main points below, and also provide specific answers to each reviewer.

**Computational complexity.** The truncated Laplacian operator is simply the trace of the Hessian matrix, where negative components are set to $0$, and no eigendecomposition is needed. Therefore, the computational cost of the regularized Stein thinning is similar to the original Stein thinning. We will improve clarity of this important point in the final version of the article.

**Tuning of $\lambda$.** In our Bayesian setting, it is unfortunately not possible to tune the parameter $\lambda$, since there is not metric to assess the quality of the samples obtained over a grid of $\lambda$ values. This is the exact same limitation encountered in practice for the bandwidth of IMQ kernel $\ell$, for which, to the best of our knowledge, there is no satisfying and systematic tuning procedure. However, as opposed to the bandwidth parameter $\ell$, we are able to provide a simple heuristic for $\lambda$, directly guided by the careful analysis of Theorem 3.5. In fact, it turns out that setting $\lambda$ to $1/m$ is consistent in practice, and we show in various experiments that this heuristic is highly efficient. Additionally, we provide two theoretical guarantees for this setting of $\lambda$: the algorithm converges (Theorem 3.5), and the bias for samples truly sampled from the target does not increase with this regularization. Notice that Figure 6 in the Supplementary Material provides experiment results for lower and higher values of $\lambda$ than $1/m$: performance strongly decreases in both cases. Besides, the goal of Theorem 3.2 is to show that the regularized Stein thinning is now sensitive to the weights of multimodal distributions, as opposed to the original Stein thinning. However, we cannot theoretically determine which exact range of values of $\lambda$ lead to good thinned samples in finite sample regimes. As suggested by the reviewers, we will comment more on the limitation of this aspect of the L-KSD in the paper.

**IMQ kernel for Theorem 2.4.** The most efficient kernel for KSD-based algorithm is the IMQ kernel, as often stated in the literature, since using other types of kernels strongly degrades performance (Riabiz et al., 2022; Chen et al., 2018). Therefore, it was of utmost importance
that our theoretical results hold for the IMQ kernel used in practice. In our opinion, it was of secondary importance to extend the analysis to radial kernels. In the case of Theorem 2.3, it happens that the result holds for radial kernels without any additional assumption than for the IMQ kernel case. We thus decided to state the result in full generality. On the contrary, extending Theorem 2.4 to radial kernels requires intricate additional assumptions about the kernel properties. More importantly, it will strongly increase the complexity of the inequality, which relates the sample size m and the threshold s, leading to an obscure and unintuitive condition. For the sake of clarity and the practical scope of the theory, we thus believe that Theorem 2.4 should only be stated for IMQ kernels.

**Additional details in the theoretical analysis.** We will take advantage of the additional page, if the paper is accepted, to add details about assumptions of the theoretical results, to improve clarity. We separately answer to the points raised by the reviewers in dedicated posts.


Riabiz, M., Chen, W. Y., Cockayne, J., Swietach, P., Niederer, S. A., Mackey, L., & Oates, C. J. (2022). Optimal thinning of MCMC output. Journal of the Royal Statistical Society Series B: Statistical Methodology, 84(4), 1059-1081.

Chen, W. Y., Mackey, L., Gorham, J., Briol, F. X., & Oates, C. (2018, July). Stein points. In International Conference on Machine Learning (pp. 844-853). PMLR.

---

### Decision · Program_Chairs · 2023-09-21

**Decision:**

Accept (poster)

**Comment:**

All reviewers agreed that this work was above the acceptance threshold for NeurIPS.  The methodology is sound and supported by appropriate theoretical analysis.  I slightly question whether the specific methodological innovations were too focused on fixing specific examples where the original Stein thinning method performed poorly, and that in making these modifications to the algorithm the authors may have introduced a different collection of examples where the method performs poorly.  But nevertheless this is a very welcome contribution to NeurIPS!